# Simple autonomous agents can enhance creative semantic discovery by human groups

Atsushi Ueshima [1,2,3,4], Matthew I. Jones [1,2,5] & Nicholas A. Christakis [1,2,6] ✉

Innovation is challenging, and theory and experiments indicate that groups may be better able to identify and preserve innovations than individuals. But innovation within groups faces its own challenges, including groupthink and truncated diffusion. We performed experiments involving a game in which people search for ideas in various conditions: alone, in networked social groups, or in networked groups featuring autonomous agents (bots). The objective was to search a semantic space of 20,000 nouns with defined similarities for an arbitrary noun with the highest point value. Participants ($N = 1875$) were embedded in networks ($n = 125$) of 15 nodes to which we sometimes added 2 bots. The bots had 3 possible strategies: they shared a random noun generated by their immediate neighbors, or a noun most similar from among those identified, or a noun least similar. We first confirm that groups are better able to explore a semantic space than isolated individuals. Then we show that when bots that share the most similar noun operate in groups facing a semantic space that is relatively easy to navigate, group performance is superior. Simple autonomous agents with interpretable behavior can affect the capacity for creative discovery of human groups.

The discovery of innovative ideas can enhance the immediate welfare of a population and even modify the course of evolution[1–3]. However, finding such valuable ideas often involves exploring a large pool of possibilities – which can be a challenging process for both individuals and groups. The primary roadblock to finding good ideas is normally not that innovations are hard to evaluate, but that coming up with an original, paradigm-shifting idea that pushes the boundary of the space of available ideas is difficult. Ironically, this is a challenge that being in groups can both mitigate and amplify. Moreover, since simple autonomous agents can alter group behavior in a variety of ways[4–8], such agents might also affect the creative capacity of groups.

For the emergence of collective intelligence, prior work has highlighted the importance of both independence and inter-dependence among group members[9–13]. The presence of too much inter-dependence within a group can result in a quick convergence on

an inferior idea (e.g., groupthink[14]). Such social herding has been shown to have negative effects on collective intelligence[15,16]. On the other hand, if there is not a focused group whose members draw inspiration from each other, the lack of inter-dependence can lead to uncoordinated and inefficient exploration of ideas and a failure to exploit any beneficial innovations once they are discovered.

Prior work on social learning within human groups has focused on critical factors including, for example, network structure[17–20], learning strategy[21–24], and group size[25–27]. However, prior experimental studies of networked collective decision-making have generally neglected the critical issue of relationships among candidate ideas – for instance, semantic similarity between ideas in an idea space. In daily life, similar ideas tend to have similar value and also tend to be easier to discover via marginal improvements to existing ideas. Groups can follow a strategy whereby members use the ideas proposed by their neighbors

[1]Yale Institute for Network Science, Yale University, New Haven, CT, USA. [2]Department of Sociology, Yale University, New Haven, CT, USA. [3]Japan Society for the Promotion of Science, Tokyo, Japan. [4]Department of Human Sciences, Faculty of Letters, Keio University, Tokyo, Japan. [5]Sunwater Institute, North Bethesda, MD, USA. [6]Department of Statistics and Data Science, Yale University, New Haven, CT, USA. ✉e-mail: nicholas.christakis@yale.edu

to help guide their next attempt. It is, therefore, important to understand the strategies that groups can adopt in order to enhance collective creativity in such a situation.

The interplay between the independence and inter-dependence in idea sharing can also inform the development of intervention strategies. For instance, a group producing overly similar ideas could benefit from an intervention that promotes independence in idea generation, thus reducing idea similarity and facilitating the discovery of novel ideas, while a group that is already effectively exploring solutions might benefit from additional sharing of ideas to promote exploitation of high-value regions of the semantic space.

Here, we first develop a word search game mimicking such challenges. Then, we test it in groups of isolated individuals and in groups that can share information in a social network; and we show that social information helps groups explore the idea space. Finally, we demonstrate how the use of simple autonomous agents (bots) can affect collective idea exploration. We test several different potential group-level interventions involving such simple bots. We also explore the impact of making the problem harder to solve by adding a variety of decoys to the idea landscape.

Despite the ongoing transformation of social and computational science research by large language models[28], here we focus on simple autonomous agents that work with classic natural language processing techniques and that are thus relatively transparent in nature[4,29]. Doing so allows us to have full control over how our AI-bots intervene in human groups; to obtain more interpretability in what the bots are doing; and to focus on human creativity rather than AI capability per se. Nevertheless, this methodology also sheds light on how more complex forms of AI might shape the behavior of human groups.

In total, we show that adding simple bots to networked human groups has a notable impact on the ability of groups to find rewarding regions of semantic space, particularly when sharing similar ideas in less challenging landscapes.

## Results
### Methodology summary
We use words (specifically, nouns) as an analogue for ideas. Both words and ideas have semantic relationships determining how similar one word or idea is to another, and both require a level of originality to come up with new and interesting examples. We performed experiments where participants were asked to search for nouns from a set of 20,000 frequently used English nouns to which we assigned arbitrary values (see Supplementary Methods for details). To incorporate the real-world aspect of similar ideas having comparable value, we used a

simple and well understood natural language processing resource known as word2vec[30,31] which is an embedding of words into a 300-dimensional space where a word's semantic meaning is conveyed by the position of its vector representation[32]. Semantically similar words such as "dog" and "cat" will have vectors with high cosine similarity, whereas dissimilar words such as "dog" and "desk" will not. Using these vectors, we are able to assign comparable point values to semantically similar nouns.

Participants were incentivized to find a single target noun which was assigned to have the highest point value of 20,000. Other nouns were then assigned points relative to their closeness to the target. For instance, if "dog" was selected as the target with a point value of 20,000, "cat" would also receive a high point value. This task replicates the many human decision-making endeavors where the options are not easily enumerable but are nevertheless related and easily evaluable (e.g., studio executives predicting how the public will react to movies, curriculum committees deciding which new classes to offer, families deciding on the best vacation, etc.). We chose a set of 18 target nouns that were spread out in the word2vec space and were roughly equally obscure (see Supplementary Methods for details), as follows: "recce", "cartography", "investiture", "comedown", "hesitance", "decile", "shoehorn", "edutainment", "narrowness", "activewear", "epee", "doyenne", "actuation", "sarcoma", "braggadocio", "jowl", "fratricide", and "translocation." In our experiments, individuals worked together to find these targets. See Fig. 1a, b for two examples of trajectories through the noun space when people were searching for "fratricide."

We report results for 1875 participants (recruited through Amazon Mechanical Turk) who were placed into 125 groups (see Supplementary Methods regarding additional participants in one of the experimental arms). Each trial involved 15 participants placed within a networked group playing five word-search games. In each game, the participants are embedded in a social network with links formed following the Erdos-Rényi model[5,33]. Each game lasted 25 rounds, and, in each round, participants were asked to submit a noun. Following each round, the point value of the noun they chose was displayed (assuming the noun was included in the list of the 20,000 nouns). Additionally, participants were provided with the most recent responses (nouns) and respective point values from their network neighbors with whom they were in direct connection (i.e., social information). To reduce mental load, participants were also shown the highest-scoring noun they had seen so far, along with its point value. Each group of 15 people played five sequential conditions involving bots, as discussed below (i.e., five games of 25 rounds), with different target nouns carrying the highest point values in each condition. To avoid order effects, the

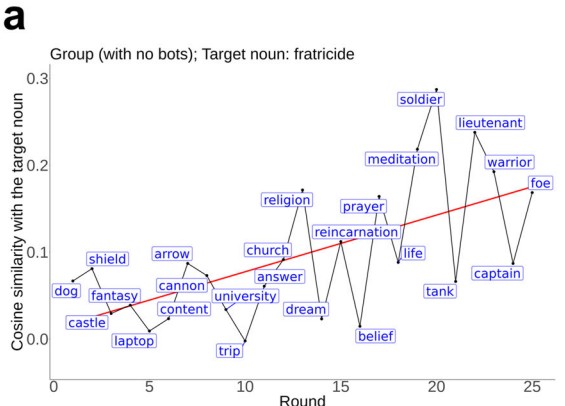

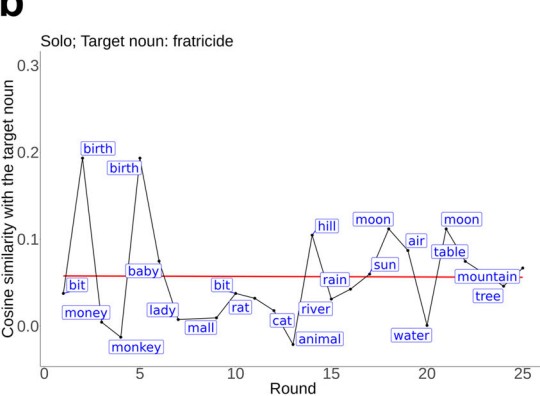

**Fig. 1 | Example trajectories of participants. a** An example answer trajectory of a participant who played very well (in the no-bot group condition). The landscape was the no-decoy condition, and the target noun was "fratricide". **b** An example answer trajectory of a participant who did not play very well (in the solo condition)

with the same landscape condition and target noun (two nouns that were not in the list of 20,000 nouns are not shown, from rounds 8 and 23). In panels a and b, best-fit lines are included in red for illustrative purposes.

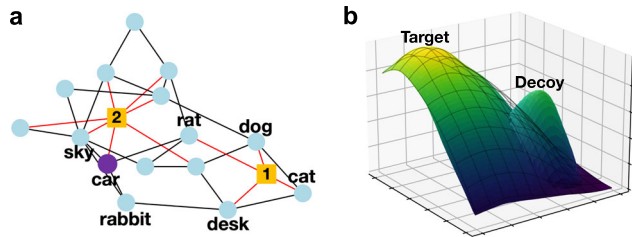

**Fig. 2 | Illustration of the bot interventions and the semantic landscape. a** An example social network consisting of 15 human players (light blue and purple circles) and two bots (yellow squares). Edges connected to a bot are colored red. Only the example nouns answered by some participants are shown. **b** A demonstration of the boosting algorithm on a mesh grid of 10,000 points in two dimensions. The black mesh shows the no-decoy landscape, where a target point is located at (0.2, 0.2) and rank is a monotone function of distance to the target. In landscapes with a decoy, the values of points close to the decoy at (0.7, 0.7) were artificially boosted, but the total rank sum of all 10,000 points was unchanged (see Supplementary Fig. 2). Higher point values are associated with light colors, while lower point values are associated with dark colors.

order of bot treatments was delivered based on a fractional factorial design (see Supplementary Methods for details).

We used autonomous agents (bots) to both explore what sort of strategies humans might naturally use that could affect the performance of their groups, and also to assess possible strategies that bots could themselves adopt to enhance the creativity of human groups. Hence, two bots were sometimes added to the human networks, resulting in a 17-node hybrid system of humans and bots[4] (Fig. 2a and Supplementary Fig. 1). In some treatment conditions, these bots used the word2vec database to assess how similar the nouns the humans offered were to each other. The participants in the network were not informed if their neighbor was a human or bot, and the social information was shown to each participant without revealing which of their neighbors offered which noun (Table 1).

There were three types of bots: the least-similar bot, the most-similar bot, and the random bot. The least-similar bot used semantic similarities to choose the word that was least similar among all its neighbors' guesses on the most recent round. For instance, if the bot's neighbors' answers during a round were "dog," "cat," "rat," and "desk," as shown in Fig. 2a (where the bot is labeled 1), the least-similar bot would select "desk" as the least-similar noun among its neighbors (Table 2). After selecting "desk" as the least-similar noun, the bot would send this word to the other bot (node labeled 2 in Fig. 2a) which could share that word with its neighbors in that same round. In other words, each bot had the ability to immediately propagate ideas received from the other bot to other (distant) regions of the networked group. Notice that each of the two bots in a network had only one candidate noun to share with its own neighbors—that is, the noun sent from the other bot. As the names suggest, the most-similar bots chose the noun that was most similar among all the guesses and the random bots chose randomly from its neighbors' responses. To be clear, the most-similar and the least-similar bots did not use information regarding the target noun to choose the noun to broadcast to another local region of the network group. In other words, the most-similar and least-similar nouns were determined independently from the target noun based solely on human participants' ideas (Table 2). Notably, the random bot could act without any information about the noun similarities at all. For the most frequently observed nouns offered by humans in the experiment, see Supplementary Tables 2 and 3.

Rugged solution landscapes can add additional difficulty when participants are searching for optimal solutions due the possibility of getting stuck on a suboptimal local maximum[4,11]. To simulate this challenge, we sometimes employed decoy nouns to potentially mislead participants. The point values of nouns semantically similar to the

**Table 1 | An example of the social information shown to a participant located at the purple node in the least-similar bot condition after the round depicted in Fig. 2a[a]**

| Player | Latest answer | Points |
|---|---|---|
| You | car | 5492 |
| Neighbor | rat | 12,293 |
| Neighbor | rabbit | 12,114 |
| Neighbor | sky | 18,999 |
| Neighbor | desk | 6709 |

[a]During the game, participants could see the information about the highest-point-value noun in a game with the description such as "The highest-point-value noun answered by you or your neighbors so far in this game: sky with 18,999 points."

**Table 2 | The calculation of pairwise cosine similarities of four example nouns observed by bot #1, which were obtained using word2vec[a]**

|  | cat | dog | rat | desk |
|---|---|---|---|---|
| cat | 1 | 0.76 | 0.53 | 0.16 |
| dog | 0.76 | 1 | 0.44 | 0.12 |
| rat | 0.53 | 0.44 | 1 | 0.06 |
| desk | 0.16 | 0.12 | 0.06 | 1 |

[a]The least or most similar noun can be determined by computing the average of the values in the rows and choosing the noun with the smallest or largest values, respectively. In this example, the least similar noun is desk. The cosine similarities with each of the other nouns are 0.16, 0.12, and 0.06. On the other hand, the most similar noun is cat. The cosine similarities with each of the other nouns are 0.76, 0.53, and 0.16.

arbitrarily chosen decoy nouns were boosted (Fig. 2b; see Supplementary Fig. 2 for details). Because the point value of the decoy noun was kept lower than the target noun value, it functioned as a local optimum in the semantic space.

We considered two parameters when implementing the decoy nouns: (1) the point value of the decoy noun (i.e., the height of the decoy peak), and (2) the number of surrounding nouns that were boosted around the decoy noun (i.e., the width of the decoy peak). We tested two variations of each parameter (tall and short, and wide and narrow) for a total of five landscapes: (1) tall/wide, (2) tall/narrow, (3) short/wide, (4) short/narrow, and (5) no decoy. The parameters were chosen so that the tall/narrow and short/wide landscapes had the same theoretical probability of misleading participants in the early stages of the game (see Supplementary Methods and Supplementary Fig. 2 for illustrations and details). Participants were not informed of the presence of the decoy nouns.

Each group participated in five games, including one for each bot treatment: least-similar bot, most-similar bot, random bot, no bot, and solo condition. In the no-bot condition, 15 participants played a game without the two bots embedded in the network, where the edges connecting the bots to the humans were removed. In the solo condition, participants played alone without any neighbors, where all the edges in the network were removed. Each group experienced the five games under the same type of decoy landscape, and each landscape therefore contained 25 unique groups. Thus, the decoy landscapes were between-participant treatments, while the bot conditions were within-participant treatments.

**Creative thinking in groups benefits from sharing similar ideas**
We begin our investigation with the contrast between the solo and group conditions. To measure group creativity, we take the average cosine similarity to the target noun over all guesses by all participants over all 25 rounds of each game. This value, which is typically small but

positive, is the dependent variable in all analyses (unless otherwise noted). Our results from a regression model using the no-bot condition as the reference variable show that working as a group adds a significant benefit over members acting in isolation ($\beta_{Solo} = -0.70$; 95% highest density interval (HDI) [−1.10, −0.33]; see the no-decoy facet in Fig. 3a), emphasizing the value of social learning in efficiently exploring semantic space. Then, in general, we find that the addition of any kind of bot (i.e., most-similar, least-similar, or random) did not yield meaningful main effects in comparison to the no-bot group situation (i.e., the red, blue, and gray lines are not statistically distinguishable from the black lines in Fig. 3a, in the no-decoy condition).

Next, we investigated the interactions between bot conditions and decoy landscapes with regression models, using the bot condition (reference variable: no bot), landscape type (reference variable: short/wide), and their interactions as the key independent variables (Fig. 3b). For groups with the most-similar bot and some of the landscapes, we found a specific performance increase that cannot be explained by either the bot type or the landscape alone ($\beta_{Most:no\ decoy} = 0.56$; 95% HDI [0.05, 1.07]; $\beta_{Most:tall/narrow} = 0.50$; 95% HDI [0.00, 1.03]). Similar trends were also shown in the short/narrow landscape ($\beta_{Most:short/narrow} = 0.44$; 95% HDI [−0.08, 0.97]; 90% HDI [0.00, 0.87]). All the 90% HDIs of other parameters included zero, demonstrating that none of the other interaction effects were statistically meaningful. There was no main effect of bot types (Fig. 3b), showing that the positive effect of the most-similar bot was not observed in comparison to the no-bot condition but was only seen as an interaction effect between bot function and landscape type. Therefore, adding the most-similar bots in a social network is most advantageous in easier landscapes with fewer artificially boosted nouns, specifically in the no-decoy and narrow landscapes.

An additional analysis did not find statistically meaningful differences (at the 95% HDI criterion) in task performance between participants who were directly connected with a bot and those who were not, in any of the bot treatments or landscapes (for detailed results, see Supplementary Fig. 3), hinting that the effects of bots were not limited to any subgroup of the network[4].

To understand why the most-similar bot had a positive effect (in certain landscapes), we assessed the quality of nouns shared by the different types of bots (i.e., the most-similar, least-similar, and random bots) in each game. Figure 4a illustrates that there was a sharing of higher quality nouns that cannot be explained by the bot type and the landscape alone for the most-similar bots employed with the easier (no-decoy, short/narrow, and tall/narrow) landscapes, compared to the least-similar bot ($\beta_{Least:no\ decoy} = -0.85$, 95% HDI [−1.42, −0.25]; $\beta_{Least:short/narrow} = -0.81$, 95% HDI [−1.45, −0.20]; $\beta_{Least:tall/narrow} = -0.81$, 95% HDI [−1.40, −0.20]; using the tall/wide landscape as the reference).

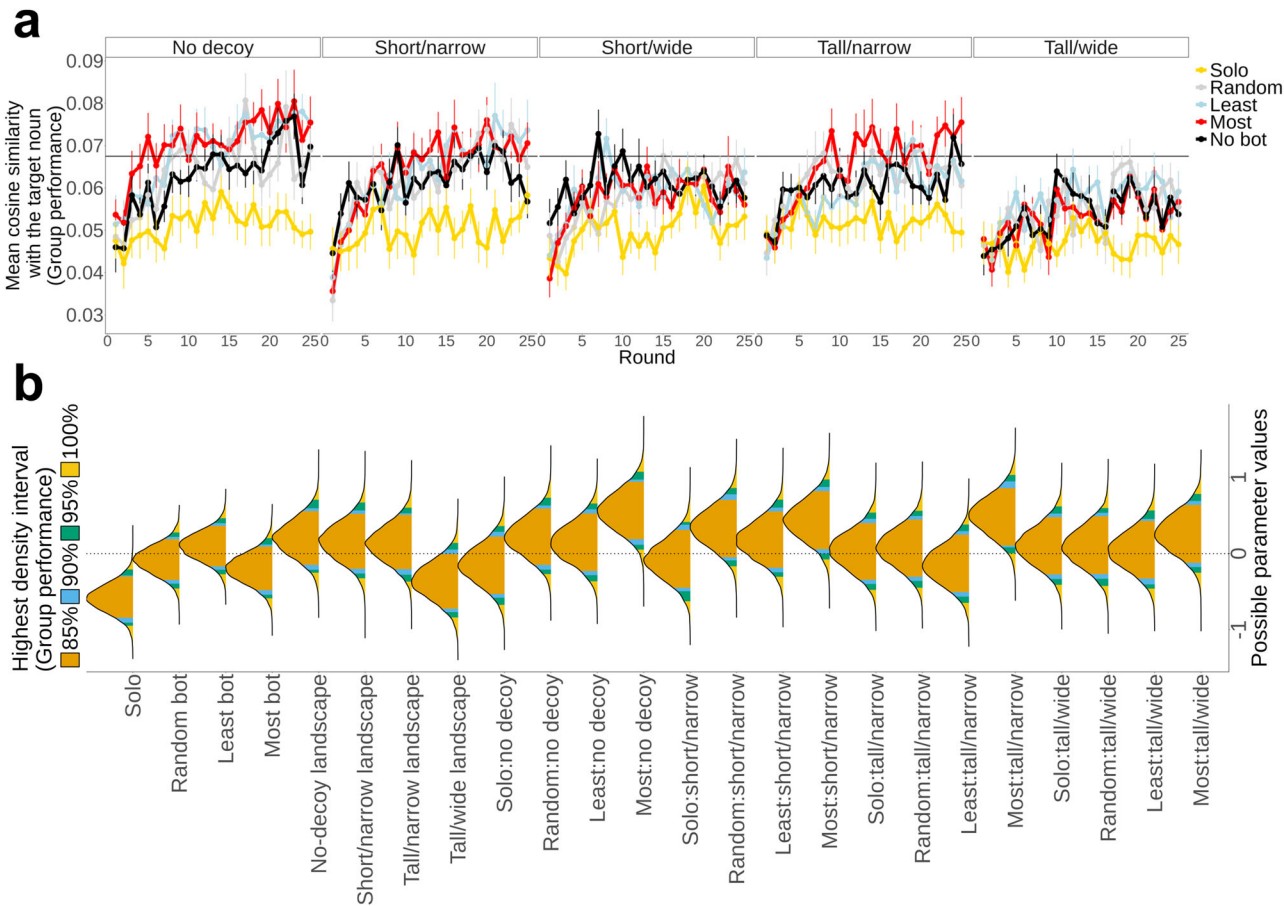

**Fig. 3 | The most-similar bot helped groups achieve better results in landscapes with fewer artificially boosted nouns. a** Mean cosine similarity between participants' answer and the target noun across 25 rounds in 5 decoy landscapes. The horizontal line indicates the mean cosine similarity between each of the 18 target nouns and the 20,000 nouns, across conditions. It is apparent that the solo condition (yellow) has the worst performance and the no-bot social condition (black) is an improvement across all landscapes. The bot conditions involving the most-similar bots (red) are helpful, especially so in the narrow landscapes. Error bars indicate standard errors. **b** Posterior distributions of regression coefficients with the computed highest density intervals. For the dependent variable of the regression analysis, we averaged the cosine similarity between answers and the target for each game. The study incorporated 125 unique groups, each completing 5 games, resulting in 625 data points. The regression model's independent variables included fixed effects of bot conditions, landscape variables, and their interactions, with the reference variables being the no-bot condition and short/wide landscape.

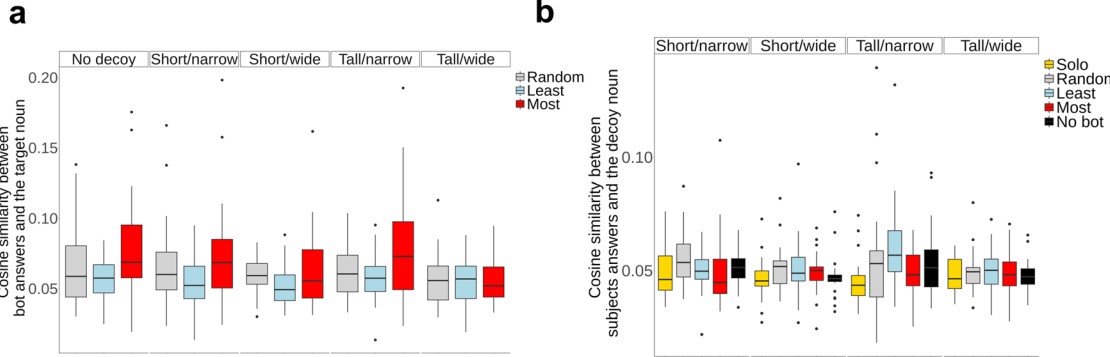

**Fig. 4 | Impact of bot behavior and decoy landscape on similarity of answers to the target noun and to the decoy noun. a** Cosine similarity between nouns shared by bots and the target noun in each game. A total of 125 groups went through three games under different bot conditions for a total of 375 data points. The regression model's independent variables included fixed effects for the three bot conditions (random, least, most), the landscape variables, and their interactions, with the reference variables being the most-similar bot condition and the tall/wide landscape. The solo and no-bot conditions were excluded because these conditions did not have bots. We see that the most similar bot (red) helped participants identify high-value nouns by propagating them through the network. **b** Cosine similarity between nouns guessed by participants and the decoy noun. The data from the no-decoy landscape was excluded, resulting in 500 data points. The regression model's independent variables included fixed effects of bot conditions, landscape variables, and their interactions, with the reference variables being the most-similar bot condition and short/wide landscapes. The most-similar bot (red) did not pull the group toward the decoy local optimum in the same way that it moved the group toward the global maximum at the target. The plots show summary statistics of the raw data presented with box-and-whisker plots. The box represents the interquartile range (IQR). The line within the box represents the median value. The upper (lower) whisker extends from the hinge to the largest (smallest) value no further than 1.5 * IQR from the hinge. Data outside the whiskers are plotted individually.

Likewise, the most-similar bots meaningfully outperform the random bots in the tall/narrow landscape ($\beta_{\text{Random:tall/narrow}} = -0.85$, 95% HDI [−1.44, −0.23]). A similar trend was also observed in in the no-decoy landscape ($\beta_{\text{Random:no decoy}} = -0.47$, 95% HDI [−1.04, 0.14]; 90% HDI [−0.98, 0.01]). We did not observe statistically distinguishable main effects of the bot types.

Although this evidence is indirect, these results suggest that the most-similar bot may have been able to help participants generate nouns that were similar to the target by propagating high-value nouns throughout a network. Importantly, this was possible because nouns relayed by the most-similar bot – i.e., the idea that was most similar among the various ideas offered by its neighbors – were indeed similar to the target noun, at least in the no-decoy and the two narrow landscapes. Humans seem to have some intrinsic ability to solve the game on these easier landscapes, and the bots may amplify this ability by leveraging the wisdom of crowds, essentially reducing noise.

Our data indicate that the addition of the most-similar bot helps groups approach the target noun, but it is conceivable that the bot could also lead groups towards the decoy noun by sharing nouns whose point values have been artificially inflated due to proximity with the target noun. To test this, we considered the similarity between the nouns offered by participants and the decoy noun in all landscapes (except the no-decoy landscape). As shown in Fig. 4b, there is no evidence that the use of the most-similar bot increased the similarity between participants' answers and the decoy noun (for detailed results, see Supplementary Fig. 4). We did not observe a meaningful main effect of the bot types, suggesting that none of the bot types led participants to the decoy when controlling for the landscape factor. On the other hand, the same regression model revealed interaction effects between the bot conditions and the landscapes; in the tall/narrow landscape, participants' answers resembled the decoy noun more with the least-similar bot and the random bot, in comparison to when they played a game with the most-similar bot ($\beta_{\text{Least:tall/narrow}} = 0.67$, 95% HDI [0.05, 1.31]; $\beta_{\text{Random:tall/narrow}} = 0.65$, 95% HDI [0.02, 1.26]; with most-similar bot and short/wide landscape used as reference variables). Therefore, the actions of the most-similar bot, sharing similar ideas from among the various ideas offered by its neighbors, are still effective at improving performance, even in the presence of local optima within the semantic landscape.

## Wider decoy peaks hindered the establishment of a semantic alignment

So far, we have found that the most-similar bot was more useful in narrow landscapes than in wide ones. This is part of a broader trend (regardless of bot behaviors) that wide landscapes posed a greater challenge for participants than narrow ones (Fig. 3a). Still, participants were indeed capable of finding nouns similar to the target noun in the wide landscapes. The maximum similarities between the nouns and the target nouns achieved in each game did not show any meaningful difference across the landscapes, as seen in Supplementary Fig. 5. However, participants were unable to take advantage of these occasional discoveries to improve their collective guesses. We attribute these findings to wider landscapes making it difficult for participants to recognize rewarding nouns as forming a meaningful cluster. In the wide landscapes, we artificially boosted the ranks of a large number of nouns (12,000) around the decoy, disrupting the correlation between semantic meaning and point value. As a result, it may have been more challenging for participants to form an accurate understanding of the fitness landscape. For example, under a coherent alignment between noun meanings and point values, when two semantically separate words like "chainsaw" and "grammar" have drastically different point values, this is a clear signal for participants to either try nouns that are tools or nouns related to language. However, if they both have high value because one is near the target and the other is near the decoy, for instance, it would be difficult for players to determine which area of the semantic landscape to continue to explore.

Based on the preceding argument, we hypothesized that individuals might exhibit distinct behavior depending on the landscape they faced. Specifically, we suspected that participants in narrower landscapes could establish cognitive alignment between point values and semantic meanings associated with specific nouns, and that, after receiving high point values in the previous round, participants would tend to answer a noun that was semantically similar to their previously observed noun. To test this possibility, we examined the correlation between point values obtained in each round $t$ and the cosine similarity between nouns in rounds $t$ and $t+1$ (Fig. 5). We specified a regression model with the main effects of the landscape width (wide, narrow, or no-decoy landscape; with the narrow one as the reference category), height (tall, short, or no-decoy landscape), and bot types. To make the

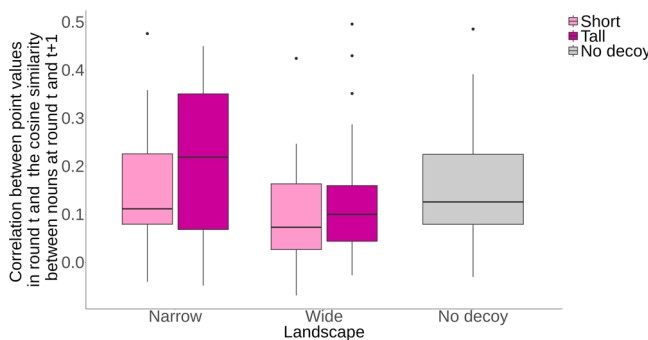

**Fig. 5 | Mean correlation between point values in round _t_ and the cosine similarity between nouns at round _t_ and _t_ + 1 in the five landscapes.** In the regression model, we controlled for the height (i.e., tall, short or no-decoy) and bot conditions to estimate the effect of width (wide, narrow, or no-decoy). The results indicate that the wider landscapes made the relationship between semantic meaning and point value less perceptible to participants. For the box-and-whisker plots, the box represents the interquartile range (IQR). The line within the box represents the median value. The upper (lower) whisker extends from the hinge to the largest (smallest) value no further than 1.5 * IQR from the hinge. Data outside the whiskers are plotted individually. Each of the boxes contains 25 independent groups (i.e., data points).

effect of the no-decoy landscape identifiable, we fixed the coefficient of the no-decoy condition in the height variable to be a constant zero and only estimate the no-decoy effect with the width variable. The results show that, regardless of the landscape, participants in the solo condition showed a lower correlation than those in the no-bot condition ($\beta_{\text{Solo}} = -0.33$, 95% HDI [−0.52, −0.16]), meaning that social information helps individuals build an accurate picture of the landscape and to make better guesses. More interestingly (and confirming our hypothesis), the model also revealed that participants in the narrower landscapes had a higher correlation than those in the wider landscapes ($\beta_{\text{Wide}} = -0.34$, 95% HDI [−0.62, −0.05]), in which players encountering a high-value noun are unable (or unwilling) to exploit that high value by guessing other nearby nouns. No significant difference was observed between the narrow landscapes and the no-decoy landscape ($\beta_{\text{No decoy}} = -0.06$, 95% HDI [−0.45, 0.35]). See Supplementary Fig. 6 for details.

These findings suggest that the relationship between semantic meaning and point value was indeed less perceptible to participants in the wider landscapes. It was not how high the decoy nouns were boosted, but rather the number of artificially boosted nouns around the decoy nouns that made a semantic space less navigable for participants. The wide decoy peaks do not attract the participants toward the local maximum; instead, they appear to scramble the word ranks so much that participants lose the ability to form a coherent (even if implicit) image of the landscape in their minds. The point values may seem indecipherable, leaving the participants struggling to decide which region of semantic space to explore.

### Solitary creativity is different than creativity in a group

To provide a deeper understanding of the mechanism underlying group creativity, we also explored the relationship between creative performance exhibited by individuals in a solitary context versus in a group context. We analyzed whether the participants who performed well in the solo condition also had successful individual performances in the other groupwise conditions. As depicted in the results shown in the top row of Supplementary Fig. 7a, the participants who could answer nouns that were similar to the target noun in the solo condition were also more likely to do so in the four social conditions, compared to those who were unable to provide good ideas. In other words, some individuals are naturally better at coming up with new nouns closer to a target. However, noticeably, this performance correlation was weak

(Pearson $r = 0.18$ to 0.23), compared to the performance correlations between social conditions (Pearson $r = 0.38$ to 0.54). In other words, success in the solo condition is a (relatively) poor indicator of individual success in a group setting. Previous work has shown that collective intelligence can be a quality exhibited by a group, not always reducible to the individuals within it[34]. Our results suggest that there might be two traits that individuals possess to some degree: solo creativity and group creativity. Some individuals have high solo creativity and low group creativity, while others have high group creativity and low solo creativity. Together with previous literature, our results therefore help shed light on why individual traits do not always explain group performance.

To further elucidate the weak correlation between the creativity of participants in solo and group conditions, we conducted an exploratory analysis focusing on two different aspects of each participant. First, we analyzed the number of unique nouns answered by each participant in each game, assuming this variable indicates their ability or motivation to come up with different nouns. Second, we analyzed the extent of divergent thinking in each participant by calculating the cosine similarity between nouns answered in rounds $t$ and $t + 1$. A higher similarity indicates a narrow exploration of a semantic space, while a lower similarity indicates wider exploration at each step. Surprisingly, when using these two aspects of a participants' tactics or abilities, the correlations between the solo and social conditions were not meaningfully different (Supplementary Figs. 7b, c), unlike the performance correlations in Supplementary Fig. 7a. These additional analyses suggest that while players come up with the same number of words with similar patterns in solo and social conditions, their performance changed meaningfully in the social condition. While participants were incentivized in a way that prevented them from behaving strategically (see Methods), individual participants could still have engaged in freeriding in the group condition, which may have resulted in a weaker correlation between creative performance in the solo and group conditions.

### Discussion

The human capacity for social learning in groups is enhanced by simple forms of AI, especially in situations where there are distractions or challenges of a certain kind. These simple bots had a notable effect on the ability of groups to find and exploit rewarding regions of semantic space, particularly when sharing similar ideas. Importantly, these bots implement a low-cost, straightforward, and decentralized algorithm, functioning solely with local neighbor information. While our bots only processed semantic embeddings of English nouns here, this approach could easily extend to non-English languages and longer sentences or more complex ideas using newer language models[35,36]. Moreover, simple autonomous agents could also be used in other settings to identify analogous beneficial behaviors used by humans themselves in human-only groups.

We observed that, when the association between semantic meaning and point value was less comprehensible to participants (in the wide decoy landscapes), groups made slower progress towards a rewarding semantic area closer to the target, nullifying the advantage obtained by the most-similar bot. The difficulties encountered in the wide landscape suggest that navigation in a semantic space may require humans to form a coherent (if implicit) image of the landscape in their minds. Certain types of AI assistance might, therefore, enhance collective intelligence by informing humans of such topical information during collective semantic navigation so that they can understand which topics consistently generate valuable ideas.

The motivation behind the current study was to investigate how a simple form of AI could affect creativity in human groups. In doing so, we focused on humans' ability to navigate through semantic space to find novel solutions. Our experiment employed a semantic search problem using a vector space model of semantic representation, given

the theoretical framework showing that semantic meanings can be represented by numeric vectors[32,37]. Moreover, we created a complex fitness landscape (i.e., decoy landscapes) to simulate optimal idea search in human collective decision-making.

Prior work has indicated that solo performers are often better at identifying the best answer because they are not vulnerable to social herding and are more likely to continue exploration[19]. In the current study, although the average guesses were better in networked groups than in collections of solo individuals (Fig. 3), we also found that groups of participants in the solo condition achieved performance that was statistically indistinguishable in terms of the best guess in the game (Supplementary Fig. 5). These results are consistent with previous literature, suggesting that the performance of solo individuals is more positively evaluated by the best solution than by the average solution. The reason participants in the solo condition did not perform better than those in the group conditions even when we looked at the best guesses might be explained by the fact that participants were required to come up with nouns and were not provided with options to explore in the present experiment. Thus, some of the positive performance shown in the group conditions should be attributed to simple heuristics that use nouns from other participants as a starting point for brainstorming rather than forming a mental model of the fitness landscape. An additional analysis confirmed that participants in the solo condition explored less than those in the other conditions (Supplementary Fig. 10a), indicating that solitary individuals had difficulties submitting nouns that were semantically distant (see also Supplementary Fig. 10b).

Prior work on the effect of network structure on collective search has demonstrated that more connected networks are more helpful for groups to converge on the appropriate solution, particularly for easy problems[17,19,20]. Consistent with this, we found that the most-similar bot, which was designed to make the network efficient in terms of both the number of edges and semantic similarities, had a positive effect in easier landscapes.

Prior research has also suggested the effectiveness of measuring creativity using natural language processing (NLP)[38]. However, until now, the main focus has been on measuring solitary individuals' creativity in a semantic space. Here, we studied social-level semantic navigation, specifically in terms of group-focused navigation through semantically rewarding domains. In this way, our findings contribute also to this field by demonstrating the effective application of NLP techniques in social learning research. Indeed, the use of an NLP resource trained on a natural corpus allowed us to incorporate the real-world semantic correlational structure into a controlled experiment by assigning similar point values to nouns with similar meanings in the real world. Retaining such naturalistic correlations that exist in the real world when designing an experimental task can enhance the generalizability of experimental findings[39,40]. Methodologically, our work thus incorporates such a paradigm in an experiment regarding human group creativity, helping to strike a balance between rigid control and generalizability.

It is noteworthy that the deployment of autonomous agents like the most-similar bot could also aggravate ideological correlations in a narrow circle of people[41]. Hence, it is essential to carefully consider the situations in which such agents can enhance human welfare. There is increasing evidence that simple (and complex) forms of AI can be added to hybrid systems of humans and machines in a beneficial way[4–6,29,42]. We emphasize that, in this experiment, it is the humans who are being creative, not the bots; the bots simply help the humans to help themselves. The bots can afford to be dumb since they are placed amidst smart humans. Such a concept could be deployed to enhance the ability of distributed online groups engaged in citizen science working together[43,44] or to help break gridlock in ideas, say, among workers in an organization (the silo phenomenon)[3] or a scientific team[45]. Of course, it seems likely that smart bots, like large language

models, when placed in hybrid systems, might have similarly complex (beneficial or detrimental) effects, and this is an area for future research[28].

The addition of bots also adds additional edges to the network (or conversely, their removal, as we have framed our approach, removes edges). Thus, some of the positive effect of the bots (compared to the no-bot condition) could be attributed to the higher connectivity or different transitivity of the bot-enabled networks compared to the human-only network (although certainly not all of the effect, since we see improvement with some bot types and not others). Therefore, the effects of bot treatments here should be considered as if they supplement an existing network instead of as if they replace current members of the network. Considering the research design in which bots (unavoidably) alter network connectivity during the game, we note that the impact of the bots employed here should be considered more than just the strict actions of the bot per se (though not when comparing across the bot treatments).

The manipulation of group size is also a promising avenue for future experiments. The potential role of size in cultural innovation has famously been evaluated in a natural experiment related to the peopling of Oceania, where island population size was associated with both tool number and complexity[46,47]. Similarly, experimental research has shown that a larger group size can enhance the adoption of valuable information[27]. On the other hand, in a set of online experiments involving information sharing in a very different context, it was shown that larger groups face greater challenges in sharing accurate information[26]. Exploring bot strategies in diverse group sizes could yield different optimal strategies for large versus small groups.

Prior research has investigated a multi-armed bandit task in which decision-makers can use spatial structure to find an optimal option[48]. One search task in prior work, for instance, leveraged actual agricultural data to connect the task to a real-world problem. Similarly, the word-search game developed for the current study allowed participants to take advantage of the real-world correlational structure by creating a controlled experimental task with corpus data. Our task extended the previously established tasks by introducing a more complex, yet tractable, search game in the context of collective decision-making in the presence of simple AI bots, thereby making the experiment well connected to a real-world problem.

The landscape that participants were asked to explore was simple in that it had only one or two peaks, but also complex, since it was a landscape over a large and high-dimensional space. We deliberately chose a simple way to create these landscapes that took into account both the simplicity and the complexity of the desired outcome. Similar to the NK model of rugged landscapes[49], our algorithm allowed us to control the level of ruggedness, but in a more bespoke way that focused on the number of words boosted and how far up the rank they moved.

In the current study, we equipped the bots with the ability to exchange their ideas with each other within the same session while the human participants in this study were not allowed to do the same. The rationale behind this design was to emulate an ability by bots to introduce common or uncommon ideas (derived from other humans themselves, elsewhere in a group) to the humans to whom the bots were locally connected. Furthermore, this allows us to maximize the impact of simple bots in human collective decision-making. Future research may explore different bot design choices.

While we adopted a mixed design (the decoy landscapes were between-participant treatments, while the bot conditions were within-participant treatments) to increase statistical power, and studied a large number of people and groups, we were still constrained by our sample size. Accordingly, we should note the danger of false discoveries[50,51], particularly for the exploratory analyses. For example, as reported, the effects of the interaction between the bot treatment and the landscape were not statistically significant when we examined

the best solution in each game (Supplementary Fig. 5) although they were with respect to the average solution (Fig. 3).

The evidence presented here suggests that adding simple bots, acting with limited (if any) knowledge, but manipulating the sharing of ideas offered by human participants themselves within broader groups, may enhance the creativity of human groups in certain circumstances. The simplicity and transparency of decision-making in such simple AI might make it more intelligible to people, thereby eliciting more trusting and sustained relationships. Simple autonomous agents, when mixed into systems of humans, might offer the same advantages as more complex and expensive ones, but with much less effort.

## Methods

### This study was approved by the Yale University Committee on the Use of Human Subjects. All ethical regulations were met in conducting the current study

Preregistration is at https://doi.org/10.17605/OSF.IO/X8GWS. The date of the preregistration was January 7, 2023. The key analyses presented in relation to Fig. 3 and Fig. 4a were preregistered analyses. Other analyses were exploratory.

### Participant recruitment

Participants were recruited through Amazon Mechanical Turk to participate in the experimental task on a website implemented using Breadboard software (available at http://breadboard.yale.edu). Prior to beginning the task, all participants gave informed consent as approved by the Yale University Committee on the Use of Human Subjects. Our tutorial contained tests designed to filter out bots and reinforce participants' knowledge about the task. Those who were unable to correctly answer all of the questions were not allowed to participate in the task. We also took other measures to detect and remove bots, including reCAPTCHA (available at https://www.google.com/recaptcha/about/) and attention checks.

Following the tutorial, participants waited up to 10 minutes for enough participants to join the experimental task in order to begin. After completing the main task, participants answered a post-session questionnaire (including basic demographics). Upon the completion of the experiment, participants received $3 compensation for participating and a bonus of up to $11 depending on their performance on the main experimental task. The duration of the whole experiment was approximately 40 minutes.

Following our pre-registered plan, we first recruited 125 groups, each consisting of 15 participants (1875 participants). This resulted in 25 unique groups for each of the five landscape variables, including the no-decoy, short/narrow, short/wide, tall/narrow, and tall/wide landscapes. Based on the initial data analysis, we decided to deviate from the pre-registered plan and recruit another 25 groups (375 participants) for the tall/wide landscape while following the same methods as in the recruitment of the original 25 groups (see Supplementary Methods for details).

### Network groups

Our network generation model began with 15 nodes and randomly generated ties between them according to the Erdos-Rényi model with a 20% tie saturation. Then, we added two nodes representing bots to the network. We randomly selected 11 of the 15 human nodes and randomly assigned 7 of them to form a tie with one of the bot nodes and the remaining 4 to form a tie with the other bot node. These numbers were chosen based on simulations which suggested that this was roughly the expected degree distribution of the two vertices with no neighbors in common that had the highest combined number of neighbors. We created a total of six different networks using this algorithm for (random) use in our experiment (see Supplementary Fig. 1 for these six networks).

### Word search game

To investigate how autonomous agents might help groups explore a semantic space, we developed a task in which participants search for high-reward nouns from a large pool of real words. Specifically, we collected 20,000 English nouns frequently used online and assigned point values or rewards to each of them, from 1 to 20,000 (see Supplementary Methods for details). We manually removed 33 vulgar or inappropriate words from the dataset so as not to be a distraction to participants in the game.

During the experiment, participants played 5 games with different nouns carrying the highest point values in each game. Each game consisted of 25 rounds, during which participants had 7 seconds to submit an answer. After a round was over, the point value of the noun each participant answered was displayed in a table on the screen for 7 seconds (Table 1). Then, the next round started, and participants could submit another noun. Participants had an additional 14 seconds during the first round of each game to familiarize themselves with the game. When a game was finished, participants were informed that the nouns may carry different point values in the next game.

Participants were also able to see the latest answers (i.e., nouns) and corresponding point values of their immediate network neighbors. In addition to this information, they could always see the noun with the highest point value so far from among those chosen by their network neighbors (or themselves) as shown in Table 1. If a participant or any of their neighbors found a noun with a higher point value than the previous highest value noun, the information was updated immediately. In the first round of a game, no social information was provided.

Participants were told that their monetary payoff from the experiment would be based on the highest point value their whole group (of 15 people) achieved in each game (the total point values from the five games were converted to dollars at the end of the experiment). They were explicitly told that even if they do not personally find the highest point noun, if someone in their group did, it would count toward their own payoff. We provided this incentive to participants based on their group's highest score to prevent any complexity that may arise from a producer-scrounger game structure[52] caused by individual-level scoring and also to ensure that there was no advantage for participants to repeat the same noun. For this reason, we alerted participants when a participant gave the same answer twice in a row within one game.

Additionally, participants were instructed that (1) among the 20,000 nouns, different nouns carried different point values, (2) nouns used in similar contexts carried similar point values, (3) when they provide a noun, they would know the point values assigned to that noun, (4) the nouns should be in lower case, singular form, and be one word, and (5) if their answer is not included in the list (of 20,000 nouns), it would be marked as invalid with zero points.

### Target and decoy nouns

We defined a target noun as the noun with the highest point value. Participants were incentivized to search for a target noun that we selected from the list of 20,000 nouns. A target noun had a point value of 20,000, and a noun that was semantically most similar to the target noun had a point value of 19,999. A noun that was semantically most distant from the target noun had a point value of 1. Thus, the point values of the 20,000 nouns ranged from 1 to 20,000. To prevent participants from making predictions about how many words were more valuable than the current guess, we multiplied the point values of each word in each game by a random number between 1 and 3 and then rounded this product to the nearest integer before displaying it on the game screen. Accordingly, the point values could exceed 20,000 depending on the multiplication factor. Thus, participants were told before the game that the highest possible point value for a noun may vary across games. For clarity, this was meant to prevent participants

from utilizing meta-strategies when playing the game. But when discussing ranks in the paper, we do not multiply them. To avoid potential biases caused by using only one target noun in the experiment, we selected 18 different nouns from different semantic clusters as target nouns and randomly used one of them in each game. See Supplementary Methods for details.

In some experimental treatments, we employed a decoy noun in addition to the target noun. The point values of the decoy noun and other nearby nouns were artificially boosted. However, the target noun was still the most valuable, with a point value of 20,000. Accordingly, the decoy noun behaved like a local optimum in the semantic space. The same 18 target nouns also functioned as decoy nouns, with each noun being assigned as the decoy for one of the other target nouns. Targets were matched with decoys in such a way that targets and decoys were semantically distant from each other (see Supplementary Methods). An illustration of the boosting algorithm can be seen in Supplementary Fig. 2. Participants did not know about the decoy.

It is noteworthy that the non-uniform distribution of nouns can add ruggedness to the idea landscape, when a lack of nouns in a region of the landscape hinders traditional hill-climbing methods. However, these pitfalls are relatively weak, particularly with the large jumps that participants make when playing this game.

### Statistical analysis and software

Analyses were conducted using the RStan v.2.21.8[53] and its interface brms v.2.19.0[54] package in R v.4.3.0[55]. In all the analyses, we used default prior distributions of the brms package and dependent variables were normalized to a mean of zero and a standard deviation of 1. To account for repeated measurements, varying intercepts for groups and varying intercepts and slopes for target nouns were included in a regression model, as indicated. In addition to the posterior mean of beta coefficients and correlation coefficients, we reported the highest density intervals (HDI) calculated by the bayestestR v.0.13.1[56] package in R. When using the Markov chain Monte Carlo (MCMC) method, the number of warm-up iterations was set to 1000, the number of post-warm-up iterations was set to 1500, and the number of chains was set to 10. In all the analyses, Rhat statistics were below 1.05, which indicates that the MCMC methods converged. We used the Python package NetworkX[57] for creating the networks.

### Reporting summary

Further information on research design is available in the Nature Portfolio Reporting Summary linked to this article.

## Data availability

The data used in this study have been deposited in the Open Science Framework repository and are available at https://doi.org/10.17605/OSF.IO/CS3R2.

## Code availability

The code used in this study has been deposited in the Open Science Framework repository and is available at https://doi.org/10.17605/OSF.IO/CS3R2.

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

## Acknowledgements

We thank Feng Fu for helpful comments on the manuscript. Wyatt Israel provided expert programming assistance, and Adam Zhang provided valuable assistance in running the experiments. This work was supported by the NOMIS Foundation (NAC), with additional support from the Pershing Square Foundation (NAC) and the Sunwater Institute (NAC), and from JSPS KAKENHI Grant Number 21J00403 (AU).

## Author contributions

Conceptualization: A.U., M.J., and N.A.C.; Methodology: A.U., M.J., and N.A.C.; Statistical and mathematical analysis: A.U. and M.J.; Funding acquisition: A.U. and N.A.C.; Supervision: N.A.C.; Writing: A.U., M.J., and N.A.C.

## Competing interests

The authors declare no competing interests.
