## [Peer Review File · Nature Communications]

Simple Autonomous Agents Affect Creative Discovery by Human GroupsReviewers' Comments:

Reviewer #1:

Remarks to the Author:

Dear Editor,

Summary of the paper:

The authors examined collective search behaviour of networked human groups through a novel word-search game. In the game, players search for rewarding words by exploring a set of 20,000 nouns that form a high-dimensional fitness landscape. The authors introduced simple "bots" playing together with human subjects, so as to assess possible strategies that bots could use to boost human creativity. They considered three different strategies of bots, namely, the least-similar, most-similar, and random bots. They also compared these conditions with no-bot groups and solo individual human participants. Another tweak in the experimental paradigm was that in some conditions there was a "decoy" noun around which the fitness was slightly boosted so that participants might be misled to the suboptimal peak. The authors systematically varied the shape of the decoy peak in two dimensions, namely, tall/short and wide/narrow.

A key finding was that groups outperformed solo individuals in discovering high-score words. However, the results of a comparison among the group conditions were mixed and seemed to interact with the shape of the decoy, where in some decoy conditions the most-similar bots could boost human performance, while in other conditions there were no significant improvement in the bot conditions compared to the no-bot condition.

The manuscript is well written and the topic must be of great interest to a wide range of scientific disciplines. However, the current form of the manuscript does not adequately provide insights into the mechanisms that might explain the observed patterns in human collective search and creative decision-making, as well as how and under what circumstances such a bot could boost human collective creativity. There are several points that the authors should address before publishing it. Please find some comments and questions below:

Comments:

Although I fully agree that it is very important to study the impact of simple bots in human decision-making, I did not understand the rationale behind the configuration of the bots they chose to implement in this study. In lines 145-149, it's mentioned that the bots could communicate with each other within the same session, but the authors did not explain the rationale behind granting such a feature to the bots. Reading the abstract, I had expected that the bots would have been allowed to behave like human subjects, so I was surprised to learn that the bots was given this ability that was not available for human participants. The authors need to give more detailed rationales behind their choice of the bots' behaviour.

I was concerned that connections between remotely-locating bots could alter the actual topology of the social network by making a bridge, and hence it would become difficult to distinguish between the effect of adding a new connection in a social network and the effect of having bots in a network.

There was another point in the bots' behaviour that the authors should give a clear explanation. In line 146, the text reads "the bot would send this word to the other bot which could share that word with its neighbours in the same round". But the "other bot" had also already chosen a single word from its own neighbours, I understand? Does this mean that bots had two candidate nouns to share, one was what chosen from its neighbours and the other one was what sent by the partner bot? Did the bots share those two nouns?

Minor comments:

I really liked the word-search task the authors developed. It is such a neat task that is simple and tractable, while being well connected to the real world problems. The task appears to be akin to other search tasks, like the spatially or conceptually correlated multi-armed bandit task (e.g., <https://www.nature.com/articles/s41562-018-0467-4>) where agents can leverage the correlation among nearby options in searching an optimal option, or the NK-problem where agents have to search over a complex fitness landscape. It would be nice if the authors can discuss how their novel task is related to such well-established paradigms.

Reviewer #2:

Remarks to the Author:

This review is for the manuscript "Simple Autonomous Agents Can Enhance Creative Discovery by Human Groups (NCOMMS-23-51211-T). Overall, my read of this manuscript was very positive: it is well-written, the research is creative and interesting, and the results are compelling.

The only real feedback that I have for the authors is a minor suggestion of integrating their motivation and results with a bit more "translational" framing — that is, how do they imagine that their results might translate into real-world decision-making and common group tasks? In my view, the research effectively explores the creativity of groups within the chosen paradigm, and demonstrate that their intervention is effective (and make some very good points along the way). However, to strengthen the theoretical foundation and enhance the practical implications of the study, I might recommend a slightly more robust connection between the motivation, experimental design, and theoretical framework.

To elaborate on this point a little: while the authors effectively present the motivation for investigating group creativity, there is an opportunity to explicitly link this motivation to real-world problem-solving and decision-making tasks. Integrating examples or scenarios from practical applications could help bridge the gap between the experimental paradigm and the broader relevance of the study. The authors may, perhaps, consider emphasizing the practical implications of the results by discussing how the paradigm itself might be extrapolated to real-world situations. This could involve discussing potential applications in organizational settings, collaborative projects, or other domains where group creativity plays a crucial role.

That is to say: the task employed itself seems to compellingly capture a certain *type* of group process or dynamic. I myself am not an expert in this area, per se, but I can imagine some pushback on the idea that the "find the target noun" paradigm might not really translate to the types of tasks and problems that groups commonly deal with out in the wild. I do appreciate that "nouns" are only a short throw from "ideas" but, in the creativity space, I find that a lot of researchers would describe or define creativity as more of an integration of two or more ideas into something completely new, rather than group members "riffing" to come to an objectively (or a priori defined) "best" or "correct" solution. Even in rather humdrum cases like hiring committees, I think that there is often a bit of vagueness in terms of what criteria is being used to identify the "best" solution.

All of this is to say: I think that the work itself is great, and I think it highlights some very, very promising directions in a very clear and well-illustrated fashion. In fact, I think that this work is likely publishable as-is and would be a valuable contribution to the literature. Really, what I'm trying to suggest is that the contribution itself may be made even greater if the authors are able to draw some more clear or explicit lines that connect the dots between the paradigm used in the study and various real-world types of examples, even if only speculatively. Essentially, this would fill in some of the gaps between theory and practice for many readers, and would hopefully shine a light on several other domains/areas that other scholars might pursue or build upon.

Sincerely,
Ryan L. Boyd

Reviewer #3:

Remarks to the Author:

The article reports on research looking at how individuals and groups search semantic spaces for targets when 'nearness' is indicated by point values and with or without the aid of bots. Participants were placed in networked groups and could see the performance of their neighbors when in the group condition. Bots that acted as connective tissue in the network and also chose the best performing word so far also increased the overall performance, especially on the easy landscapes.

The work is interesting and well-written. The semantic search is similar to some of my work, and I think the networked/group search is an innovation that I'm previously aware. Moreover, the data and results seem sound.

My main suggestion is that the work misses a number of key things in the group search literature that I think would be easy improvements. Some of the prior work I mention below is consistent with and predicts the results. The authors might want to say more about what is novel here.

1. The work needs to better reflect on the prior literature on group search. There is a substantial amount of work from Goldstone, Galesic, Hahn, Mason, and others that I feel is missing and which has important implications for how the present work is interpreted. Indeed, as I note below, many of the results follow straightforwardly from this prior work, so it seems highly relevant. I discuss this work in an upcoming book that has a chapter on group search, which can be found here:

<https://osf.io/preprints/psyarxiv/eyrzg> There is no need to cite this chapter, but it goes into detail better than I can here how the prior work of the authors above have investigated the many variations of group search and how network structure matters in those contexts. I say more about some of these below.

2. One key role of bots in this work is to help the network become fully connected. By broadcasting the best (most-similar) finding so far, this is effectively like creating a more fully connected network. The bot does the job of people searching for the best outcome so far, but since we know that people sometimes do this already, does the bot offer something more than expediency? Moreover, we know from some of the past work mentioned above that for easy problems fully connected networks lead to more rapid convergence on the appropriate solution. This speaks directly to the modulation of exploration vs exploitation created by altering edge connectivity in the network.

3. Because of their connectivity, bots in this work basically act as filters sharing high or low quality information. Does this really require bots? Since bots share information and seem to make a collective decision, this broadcasts information of a certain kind. One could simply have a top level node that makes an executive decision to share high or low quality information. I agree this is a kind of simple bot, but I feel that what the bot does is more important than the label of 'bot'.

4. In much past work on group search, it is often important to look at the best solution in a group of solitary individuals, not the average solution. The average performance in a solo condition necessarily is drawn down by the limits of individual exploration. So the question is, is the best decision in the solo group different from the best decision in the social no-bot group? In many contexts, I would expect solo performers to be better at identifying the global maximum, because they maintain exploration for longer and are not drawn into social following that leads to local maxima too rapidly. This can be seen in the work of Mason and Goldstone.

5. What is a "landscape arm"?
6. Did participants know about the decoys?
7. Least-similar and most-similar is relative to the target, correct?
8. p8, line 271. It isn't "averaging" is it? Its the max or the min. Or maybe I've misunderstood.
9. p10 line 341: the finding suggests that solo individuals explore more, whereas social groups engage more in following (as found in prior work). There might be something interesting here, but it also seems somewhat intuitive that solo individuals know they must explore more to traingulate on the target. It's interesting that they do, by the way, so it's not a critique.
10. There is ample work on social following cited in the chapters provided above. It might also be good to report on group variance, etc (for example, one can subsample solo individuals at the same size as groups to create a point of comparison for groups). This would help to better compare groups with solo individuals of the same number, to better characterize the differences in how they search.

Thomas Hills

Reviewer #4:

Remarks to the Author:

The authors present a large-scale controlled experiment on the impact of bots on collective innovation in networks. In a 5x5 design of in total of 25 conditions, the impact of three different bots on the collective performance is investigated. The bot has two main functions, a) aggregate information in their local neighborhood and b) spread information across the network. The experimental design is intriguing. In particular, using word2vec as the backdrop for a fitness landscape is an exciting innovation. Yet, the authors fail to formulate a clear hypothesis in the main manuscript regarding the effects of the bots. Furthermore, the results are inconclusive. The authors claim that bots in general and the "most-bot" specifically enhance creative discovery of human groups. Unfortunately, this claim does not seem to supported by the data. The main concern here is not that results might be negative or inconclusive, but a lack of a strategy to mitigate false discoveries, and a misleading presentation of results. For this reason, I suggest revising the results section, potentially following the hypothesis outlined in the pre-registration.

The authors are addressing very relevant questions, regarding the impact of artificial agents on cultural processes with a focus on instant communication and aggregation. I would have liked to read a hypothesis of how these effects influence collective innovation. Unfortunately, although hypotheses are stated in the preregistration these are not picked up in the manuscript. Thus it is not clear if the presented analysis is hypothesis-driven. For an explorative analysis of 16 interaction terms, I would have expected some discussion on the danger of false discoveries and would have hoped for a related mitigation strategy.

There are important contradictions in the manuscript. In the abstract, one of the main presented findings is that the "most bot" enhances the exploration compared to groups without bots (27-29) and that bots, in general, enhance the capacity for creative discovery in human groups (31-32) [also: 469-471]. This is in contradiction to the first finding in the result section, which states that "the addition of any kind of bot [...] did not yield meaningful main effects". Specifically for the most bot, Figure 2.b suggests that the larger part of the posterior density is below the zero line, suggesting, if anything, an inverse relationship.

The authors then analyze interactions between landscapes and bot types. Here they find a positive effect for "most: no decoy" and the "most: tall/narrow" interaction. Based on this, the authors conclude that "Groups with the most similar bot were found to be closer to the target noun in comparison to those in the no-bot condition, particularly in the no-decoy and the tall/narrow landscapes". The first part of the sentence is in contradiction to the analysis of the main effects that clearly states that no difference between bot and non-bot conditions can be found (see previous paragraph). As the first part of the sentence is not supported, neither can the second part be supported based on the interaction alone. The interaction term does allow to conclude that for groups with the most bots on "easier" landscapes, there was a specific performance increase that cannot be explained by the bot type and the landscape alone. However, the interaction term alone does not allow to conclude a performance increase in comparison to the "no-bot condition" as the authors suggest. Such statement needs to also take into account the main effect of the bot type. In particular, as main effect (Most bot) appears to go in the opposite direction compared to the interaction (e.g. most: no decoy).

Similarly, a few paragraphs later (257-261) the text suggests a pairwise comparison of the "most" and "least" bot across different landscape types. The statistical analysis of parameters listed in brackets seems however only to reflect the interaction between bot type and landscape. Here again, the main effect between both types of bots seems to be not correctly accounted for.

Also in lines 296-301 main effects do not seem to be correctly accounted for.

The experimental design presented by the authors is an exiting innovation. It uses semantic similarity of nouns as extracted by contrastive learning (word2vec). The authors then construct an artificial fitness landscape on top of this semantic space. The word2vec space has two advantages, first it reflects in its structure a real-world concept space. As the author put, we can expect idea space to follow similar structures. Second, the word2vec space has been found to align with human associative similarity. This, as the authors have shown, allows naive participants to meaningful explore the semantic space.

Additional presented findings include:

- * Groups outperform individuals in searching on a fitness landscape
- * The most central items in the neighborhood of individual agents (here specifically bots) are of higher value than the least central one or a random one
- * The type of roughness informs the exploration pattern.
- * Individual performance only partially predicts performance in a group

The first two findings relate to the wisdom of the crowd effect. The third is novel to me. I would be curious to see a comparison with some rational learning strategy (see below). The last seems to relate to the notion of a collective intelligence factor c . I appreciate the efforts of the authors to report these findings in detail, however, the reader might benefit from stronger linking these findings to the existing literature on the wisdom of the crowd and collective intelligence.

The author analyses the correlation between the correlation of point values and the exploration distance (semantic distance between consequential nouns). They find this correlation to be larger on more narrow landscapes. The authors suggest that wider decoy peaks hinder cognitive alignment (~ 330). I would suggest exploring an alternative hypothesis. For instance, it might be rational for a Bayesian agent to explore more locally on a more pointy landscape when approaching a peak.

In conclusion, while the experimental design is innovative and the execution is impressive, some of the main claims made by the authors are in my view not sufficiently supported. Correspondingly, I suggest a thorough revision of the result section. Code and data has not been provided and was correspondingly not reviewed. As a final remark, the work could more clearly demonstrate its novelty.

Minor remarks:

Abstract: The last sentence seems to miss a connection.

Introduction: Although the impact of bots on cultural evolution appears to be the main focus of this research, bots are only mentioned in the very last two paragraphs. I suggest improving the link between the general remarks on cultural evolution and bots and clearly stating the motivation for the experimental design.

281: Figure 3a shows that the most central word within the bot neighborhood is of higher value compared to the least or a random word for most landscapes. This, however, does not speak about whether this helped subjects. I suggest more careful wording.

274: I got a bit confused by the figure description of Figure 3. Does it show the mean of the raw data or estimates of model parameters?

291-292: Something seems to be missing in this sentence, as the explanation seems to be incomplete.

263: I was surprised to see two different HDI reported (90% and 95%) for different parameters

340: Could it be that participants in the group just get inspired and use nouns from other participants as a starting point for their brainstorming? Simple heuristics without any world model might be able to explain the same behavior. I suggest being careful with concluding that participants form some kind of model of the fitness landscape.

Reviewer #1's comments

Dear Editor,

Summary of the paper:

The authors examined collective search behaviour of networked human groups through a novel word-search game. In the game, players search for rewarding words by exploring a set of 20,000 nouns that form a high-dimensional fitness landscape. The authors introduced simple "bots" playing together with human subjects, so as to assess possible strategies that bots could use to boost human creativity. They considered three different strategies of bots, namely, the least-similar, most-similar, and random bots. They also compared these conditions with no-bot groups and solo individual human participants. Another tweak in the experimental paradigm was that in some conditions there was a "decoy" noun around which the fitness was slightly boosted so that participants might be misled to the suboptimal peak. The authors systematically varied the shape of the decoy peak in two dimensions, namely, tall/short and wide/narrow.

A key finding was that groups outperformed solo individuals in discovering high-score words. However, the results of a comparison among the group conditions were mixed and seemed to interact with the shape of the decoy, where in some decoy conditions the most-similar bots could boost human performance, while in other conditions there were no significant improvement in the bot conditions compared to the no-bot condition.

The manuscript is well written and the topic must be of great interest to a wide range of scientific disciplines. However, the current form of the manuscript does not adequately provide insights into the mechanisms that might explain the observed patterns in human collective search and creative decision-making, as well as how and under what circumstances such a bot could boost human collective creativity. There are several points that the authors should address before publishing it. Please find some comments and questions below:

We are grateful for these positive comments. We have thoroughly responded to each comment raised by the reviewer as explained below. The issue of mechanisms can only be pursued to a certain depth in work with a design and focus such as ours, as we explain below.

Comments:

Although I fully agree that it is very important to study the impact of simple bots in human decision-making, I did not understand the rationale behind the configuration of the bots they chose to implement in this study. In lines 145-149, it's mentioned that the bots could communicate with each other within the same session, but the authors did not explain the rationale behind granting such a feature to the bots. Reading the abstract, I had expected that the bots would have been allowed to behave like human subjects, so I was surprised to learn that the bots was given this ability that was not available for human participants. The authors need to give more detailed rationales behind their choice of the bots' behaviour.

We apologize for the insufficient explanation regarding how the bots worked in the experiment. We did not mean to raise the expectations that bots are simply alternative kind of humans. Of course, bots can differ from humans in many respects, such as their speed, response, and ability to process or share information efficiently. They also differ from the humans in that they did not themselves generate new ideas in the present study.

We endowed our bots with the ability to exchange ideas with each other to maximize the impact of simple bots in human collective decision-making and we did explore several bot strategies. Of course, we are aware of other possible choices for the bots' behavior. We have revised the manuscript to clarify the rationale and to describe the awareness of other specifications, as follows:

(lines 559-565)

In the current study, we equipped the bots with the ability to exchange their ideas with each other within the same session while the human participants in this study were not allowed to do the same. The rationale behind this design was to emulate an ability by bots to introduce common or uncommon ideas (derived from other humans themselves, elsewhere in a group) to the humans to whom the bots were locally connected. Furthermore, this allows us to maximize the impact of simple bots in human collective decision-making. Future research may explore different bot design choices.

I was concerned that connections between remotely-locating bots could alter the actual topology of the social network by making a bridge, and hence it would become difficult to distinguish between the effect of adding a new connection in a social network and

the effect of having bots in a network.

We agree with the reviewer's comment that our bots impact the social network's topology by creating a bridge in every round. We acknowledged this design choice in the Discussion section as follows:

(lines 520-530)

The addition of bots also adds additional edges to the network (or, their removal, as we have framed our approach, removes edges). Thus, some of the positive effect of the bots (compared to the no-bot condition) could be attributed to the higher connectivity or different transitivity of the bot-enabled networks compared to the human-only network (although certainly not all of the effect, since we see improvement with some bot types and not others). Therefore, the effects of bot treatments here should be considered as if they supplement an existing network instead of as if they replace current members of the network. Considering the research design in which bots (unavoidably) alter network connectivity during the game, we note that the impact of the bots employed here should be considered more than just the strict actions of the bot per se (though not when comparing across the bot treatments).

There was another point in the bots' behaviour that the authors should give a clear explanation. In line 146, the text reads "the bot would send this word to the other bot which could share that word with its neighbours in the same round". But the "other bot" had also already chosen a single word from its own neighbours, I understand? Does this mean that bots had two candidate nouns to share, one was what chosen from its neighbours and the other one was what sent by the partner bot? Did the bots share those two nouns?

Thank you for pointing out the important methodological point. The reviewer is correct that *the "other bot" had also already chosen a single word from its own neighbours*. However, each bot had only one candidate noun to share with its own neighbors—that is, the noun shared by the *partner* bot. More concretely, if bot 1 chose the noun 'car' from its neighbors, that noun was sent to the other bot and was not a candidate for bot 1 to share with its neighbors. We have revised the manuscript to clarify this point:

(lines 156-157)

Notice that each of the two bots in a network had only one candidate noun to share with its own neighbors—that is, the noun sent from the *other* bot.

Minor comments:

I really liked the word-search task the authors developed. It is such a neat task that is simple and tractable, while being well connected to the real world problems. The task appears to akin to other search tasks, like the spatially or conceptually correlated multi-armed bandit task (e.g., <https://www.nature.com/articles/s41562-018-0467-4>) where agents can leverage the correlation among nearby options in searching an optimal option, or the NK-problem where agents have to search over a complex fitness landscape. It would be nice if the authors can discuss how their novel task is related to such well-established paradigms.

We are pleased that the reviewer liked the word-search task. We completely agree that our word-search task bears a resemblance to the multi-armed bandit task introduced by Wu et al. (2018, *Nature Human Behaviour*). In response to the reviewer's suggestion, we have included discussions on how our task relates to previous research and the NK-problem, as follows:

(lines 542-557; Note that citations were represented by numbers in the main text)
Prior research has investigated a multi-armed bandit task in which agents can use spatial structure to find an optimal option (Wu et al., 2018). One search task in prior work, for instance, leveraged actual agricultural data to connect the task to a real-world problem. Similarly, the word-search game developed for the current study allowed participants to take advantage of the real-world correlational structure by creating a controlled experimental task with corpus data. Our task extended the previously established tasks by introducing a more complex, yet tractable, search game in the context of collective decision-making in the presence of simple AI bots, thereby making the experiment well connected to a real-world problem.

The landscape that participants were asked to explore was simple in that it had only one or two peaks, but also complex, since it was a landscape over a large and high-dimensional space. We deliberately chose a simple way to create these landscapes that took into account both the simplicity and the complexity of the desired outcome. Similar to the NK model of rugged landscapes (Kauffman and Weinberger, 1989), our algorithm allowed us to control the level of ruggedness, but in a more bespoke way that focused on the number of words boosted and how far up the rank they moved.

Reviewer #2's comments

This review is for the manuscript "Simple Autonomous Agents Can Enhance Creative 1 Discovery by Human Groups (NCOMMS-23-51211-T). Overall, my read of this manuscript was very positive: it is well-written, the research is creative and interesting, and the results are compelling.

The only real feedback that I have for the authors is a minor suggestion of integrating their motivation and results with a bit more "translational" framing — that is, how do they imagine that their results might translate into real-world decision-making and common group tasks? In my view, the research effectively explores the creativity of groups within the chosen paradigm, and demonstrate that their intervention is effective (and make some very good points along the way). However, to strengthen the theoretical foundation and enhance the practical implications of the study, I might recommend a slightly more robust connection between the motivation, experimental design, and theoretical framework.

To elaborate on this point a little: while the authors effectively present the motivation for investigating group creativity, there is an opportunity to explicitly link this motivation to real-world problem-solving and decision-making tasks. Integrating examples or scenarios from practical applications could help bridge the gap between the experimental paradigm and the broader relevance of the study. The authors may, perhaps, consider emphasizing the practical implications of the results by discussing how the paradigm itself might be extrapolated to real-world situations. This could involve discussing potential applications in organizational settings, collaborative projects, or other domains where group creativity plays a crucial role.

That is to say: the task employed itself seems to compellingly capture a certain *type* of group process or dynamic. I myself am not an expert in this area, per se, but I can imagine some pushback on the idea that the "find the target noun" paradigm might not really translate to the types of tasks and problems that groups commonly deal with out in the wild. I do appreciate that "nouns" are only a short throw from "ideas" but, in the creativity space, I find that a lot of researchers would describe or define creativity as more of an integration of two or more ideas into something completely new, rather than group members "riffing" to come to an objectively (or a priori defined) "best" or

"correct" solution. Even in rather humdrum cases like hiring committees, I think that there is often a bit of vagueness in terms of what criteria is being used to identify the "best" solution.

All of this is to say: I think that the work itself is great, and I think it highlights some very, very promising directions in a very clear and well-illustrated fashion. In fact, I think that this work is likely publishable as-is and would be a valuable contribution to the literature. Really, what I'm trying to suggest is that the contribution itself may be made even greater if the authors are able to draw some more clear or explicit lines that connect the dots between the paradigm used in the study and various real-world types of examples, even if only speculatively. Essentially, this would fill in some of the gaps between theory and practice for many readers, and would hopefully shine a light on several other domains/areas that other scholars might pursue or build upon.

Sincerely,
Ryan L. Boyd

We are glad that the reviewer's reading of our manuscript was so positive. We also completely agree with the reviewer's point. We have revised the manuscript to incorporate a clearer explanation of the relationship between the motivation, experimental design, and theoretical framework, as follows:

In the Introduction:

(lines 108-112)

This task replicates the many human decision-making endeavors where the options are not easily enumerable but are nevertheless related and easily evaluable (e.g., studio executives predicting how the public will react to movies, curriculum committees deciding which new classes to offer, families deciding on the best vacation, etc.).

In the Discussion:

(lines 459-465; Note that citations were represented by numbers in the main text)

The motivation behind the current study was to investigate how a simple form of AI could affect creativity in human groups. In doing so, we focused on humans' ability to navigate through semantic space to find novel solutions. Our experiment employed a semantic search problem using a vector space model of semantic representation, given the theoretical framework showing that semantic meanings

can be represented by numeric vectors (Bhatia, 2017; Günther et al., 2019). Moreover, we created a complex fitness landscape (i.e., decoy landscapes) to simulate optimal idea search in human collective decision-making..

Reviewer #3's comments

The article reports on research looking at how individuals and groups search semantic spaces for targets when 'nearness' is indicated by point values and with or without the aid of bots. Participants were placed in networked groups and could see the performance of their neighbors when in the group condition. Bots that acted as connective tissue in the network and also chose the best performing word so far also increased the overall performance, especially on the easy landscapes.

The work is interesting and well-written. The semantic search is similar to some of my work, and I think the networked/group search is an innovation that I'm previously aware. Moreover, the data and results seem sound.

We are pleased to learn that the reviewer found our work interesting and relevant.

My main suggestion is that the work misses a number of key things in the group search literature that I think would be easy improvements. Some of the prior work I mention below is consistent with and predicts the results. The authors might want to say more about what is novel here.

We thank the reviewer for notifying us of prior literature that is quite relevant to our findings. We have added and addressed these citations. And we have addressed the points raised by the reviewer as described below to make the contributions of our work even clearer.

1. The work needs to better reflect on the prior literature on group search. There is a substantial amount of work from Goldstone, Galesic, Hahn, Mason, and others that I feel is missing and which has important implications for how the present work is interpreted. Indeed, as I note below, many of the results follow straightforwardly from this prior work, so it seems highly relevant. I discuss this work in an upcoming book that has a chapter on group search, which can be found here: <https://osf.io/preprints/psyarxiv/eyrzg> There is no need to cite this chapter, but it goes into detail better than I can here how the prior work of the authors above have investigated the many variations of group search and how network structure matters in those contexts. I say more about some of these below.

We completely agree that the reviewer's chapter and the literature it contains are very relevant. We revised the Introduction to incorporate and address the following literature suggested by the reviewer:

As a reviewer article:

Hills, T. Group Problem Solving: Harnessing the Wisdom of the Crowds. <https://osf.io/eyrzg> (2023).

On social herding:

Lorenz, J., Rauhut, H., Schweitzer, F. & Helbing, D. How social influence can undermine the wisdom of crowd effect. *Proc. Natl. Acad. Sci. U.S.A.* 108, 9020–9025 (2011).

Raafat, R. M., Chater, N. & Frith, C. Herding in humans. *Trends in Cognitive Sciences* 13, 420–428 (2009).

On the influence of network structure:

Mason, W. A., Jones, A. & Goldstone, R. L. Propagation of innovations in networked groups. *Journal of Experimental Psychology: General* 137, 422–433 (2008).

Hahn, U., Hansen, J. U. & Olsson, E. J. Truth tracking performance of social networks: how connectivity and clustering can make groups less competent. *Synthese* 197, 1511–1541 (2020).

On the influence of learning strategies:

Barkoczi, D. & Galesic, M. Social learning strategies modify the effect of network structure on group performance. *Nat Commun* 7, 13109 (2016).

Campbell, C. M., Izquierdo, E. J. & Goldstone, R. L. Partial copying and the role of diversity in social learning performance. *Collective Intelligence* 1, 263391372210818 (2022).

2. One key role of bots in this work is to help the network become fully connected. By broadcasting the best (most-similar) finding so far, this is effectively like creating a more fully connected network. The bot does the job of people searching for the best outcome so far, but since we know that people sometimes do this already, does the bot offer something more than expediency? Moreover, we know from some of the past work mentioned above that for easy problems fully connected networks lead to more rapid convergence on the appropriate solution. This speaks directly to the modulation of exploration vs exploitation created by altering edge connectivity in the network.

The reviewer raised two points here.

First, the reviewer asked if the bots offered something more than expediency in the experiment when they broadcast the best finding so far. Regarding how the bots worked in the study, please first allow us to clarify that the bots did not necessarily broadcast the best finding so far. To clarify, the least-similar and most-similar had nothing to do with the target. Please let us explain this using Fig. 1e from the manuscript.

Pairwise cosine similarities

	cat	dog	rat	desk
cat	1	0.76	0.53	0.16
dog	0.76	1	0.44	0.12
rat	0.53	0.44	1	0.06
desk	0.16	0.12	0.06	1

Least similar noun: desk (0.16, 0.12, 0.06)

Most similar noun: cat (0.76, 0.53, 0.16)

Suppose that bot #1, one of the two bots, observed these four nouns from its neighbors. The figure indicates the calculation of pairwise cosine similarities for four nouns observed by bot #1. The cosine similarities were determined based on the word vectors of each noun, obtained from the word2vec resource. The least or most similar noun can be determined by computing the average of the values in the rows and choosing the noun with the smallest or largest values, respectively. That is to say, the least-similar and most-similar nouns are determined independently from the target noun. In sum, the most-similar bot did not necessarily broadcast the closest nouns to the target. And the bots offered something more than expediency by calculating cosine similarities. We have clarified this point in the revised manuscript as follows:

(lines 159-163)

To be clear, the most-similar and the least-similar bots did not use information regarding the target noun to choose the noun to broadcast to another local region of the network group. In other words, the most-similar and least-similar nouns were determined independently from the target noun based solely on human subjects' ideas (**Fig. 1e**).

Second, however, the reviewer correctly pointed out that the bots helped the network have more topological connectivity, and that, especially in the case of the most-similar bot, which was designed to increase not only topological connectivity but semantic connectivity (or similarities), it led the network to more rapid convergence on the appropriate solution (Fig. 2a). To make a connection with past literature on this point,

we revised the Discussion section as follows:

(lines 486-490; Note that citations were represented by numbers in the main text)
Prior work on the effect of network structure on collective search has demonstrated that more connected networks are more helpful for groups to converge on the appropriate solution, particularly for easy problems (Derex and Boyd, 2016; Hahn et al., 2020; Mason et al., 2008). Consistent with this, we found that the most-similar bot, which was designed to make the network efficient in terms of both the number of edges and semantic similarities, had a positive effect in “easier” landscapes.

3. Because of their connectivity, bots in this work basically act as filters sharing high or low quality information. Does this really require bots? Since bots share information and seem to make a collective decision, this broadcasts information of a certain kind. One could simply have a top level node that makes an executive decision to share high or low quality information. I agree this is a kind of simple bot, but I feel that what the bot does is more important than the label of 'bot'.

We agree with the reviewer that it might be possible and reasonable to say that the current study implemented a certain kind of top-level nodes without mentioning the term of “bot” given that the bots not only calculated cosine similarities but alter the network connectivity during the game. At the same time, we believe that it also makes sense to use the label of “bot” as long as we clarify the bots’ function (and especially since the bots, by design, use only local knowledge, even if they afford some long-range communications). That is, a crucial issue is that we want to explore strategies that only require local knowledge and small-scale interactions, not an entity that has global knowledge of the whole system. We hope this is OK.

4. In much past work on group search, it is often important to look at the best solution in a group of solitary individuals, not the average solution. The average performance in a solo condition necessarily is drawn down by the limits of individual exploration. So the question is, is the best decision in the solo group different from the best decision in the social no-bot group? In many contexts, I would expect solo performers to be better at identifying the global maximum, because they maintain exploration for longer and are not drawn into social following that leads to local maxima too rapidly. This can be seen in the work of Mason and Goldstone.

We understand this point. Thank you for raising an important question (“Is the best

decision in the solo group different from the best decision in the social no-bot group?”). While the average guesses were better in groups than solo (Fig. 2), we found that solo performers achieved the same level of performance (i.e., statistically indistinguishable performance) in terms of the best guess in the game (please see Extended Data Fig. 5) because almost all groups had at least one guess very close to the target noun. These differences in the average and the best guesses are supportive for what the reviewer indicated. We added some additional descriptions of this point while referring to Mason, Jones, and Goldstone (2008) in the revised Discussion section, as per below.

(lines 467-472; Note that citations were represented by numbers in the main text)
Prior work has indicated that solo performers are often better at identifying the best answer because they are not subject to social herding and are more likely to continue exploration (Mason, Jones, and Goldstone, 2008). In the current study, although the average guesses were better in networked groups than in collections of solo individuals (**Fig. 2**), we also found that groups of participants in the solo condition achieved performance that was statistically indistinguishable in terms of the best guess in the game (Extended Data Fig. 5).

We also added a note as follows:

(lines 936-939)
These results indicate that this task was simple enough that all groups (regardless of treatment) had at least one guess that was very close to the target noun, so using the best guess instead of the average guess would not allow enough variability in our results to make any statistical claims about the effect of treatment on a group’s best guess.

5. What is a "landscape arm"?

A “landscape arm” simply indicates the landscape treatment. To improve readability, we removed the word “arm” in the revised manuscript.

6. Did participants know about the decoys?

No, they were not aware of the existence of the decoys. We have clarified this.

7. Least-similar and most-similar is relative to the target, correct?

We apologize for the unclear description of the bots’ function. We have revised the

manuscript as we responded to the reviewer's comment #2.

8. p8, line 271. It isn't "averaging" is it? Its the max or the min. Or maybe I've misunderstood.

Thank you for pointing this out. For clarity, we removed the words "averaging guesses" in the revised manuscript.

9. p10 line 341: the finding suggests that solo individuals explore more, whereas social groups engage more in following (as found in prior work). There might be something interesting here, but it also seems somewhat intuitive that solo individuals know they must explore more to triangulate on the target. It's interesting that they do, by the way, so it's not a critique.

We agree this is a very good point. We analyzed the cosine similarity between nouns at round t and $t+1$ from each participant for each of the bot conditions, a measure that indicates the average exploration tendency of individuals. Quite interestingly, we found that the average cosine similarity between consecutive nouns was smaller when participants were in the solo condition than when they were in the other conditions. That is, participants in the solo condition explored less than those in the other conditions (please see the plot below).

This result is apparently inconsistent with some prior literature (e.g., Mason, Jones, and Goldstone, 2008). However, this result can be considered reasonable given that participants in the current study were not provided with options to explore and needed

to come up with semantically different nouns to explore on their own, unlike previous research which presented participants with options to select. This experimental design made it challenging for the solo individuals to submit semantically more different nouns than for those in the other conditions because the solo performers could not use nouns from neighboring participants as a starting point for brainstorming.

We thank the reviewer for suggesting this analysis that allows us to link the current study to prior ones. We have added this analysis in the Discussion section with a new figure (Extended Data Fig. 10a).

10. There is ample work on social following cited in the chapters provided above. It might also be good to report on group variance, etc (for example, one can subsample solo individuals at the same size as groups to create a point of comparison for groups). This would help to better compare groups with solo individuals of the same number, to better characterize the differences in how they search.

Thomas Hills

We agree this is also a good point. First, please allow us to clarify that participants in the current study went through both the group conditions (most-similar bot, least-similar bot, random bot, and no-bot conditions) and the solo condition. Thus, we do not need to subsample solo individuals to conduct an analysis on group variance.

We conducted an additional analysis on the variance of the performance of 15 participants among the five conditions (most-similar bot, least-similar bot, random bot, no-bot, and solo condition). We first calculated the average performance of each participant in each of the five conditions across experimental groups. Then we calculated the variance of the performance for each group (5 conditions * 125 groups = 625 data points of the performance variance). Note that we focused on the average solution instead of the best solution (we discussed the best guess topic above).

As shown in the plot below, we found that the performance variance in each group was smaller when participants were in the solo condition (i.e., 15 isolated individuals) than when they were in other conditions.

This result indicates that when participants were in a solitary situation, they were poor performers in general (shown as in Fig. 2a). When they were in a group situation (most-similar bot, least-similar bot, random bot, and no-bot conditions), some participants may have been able to get inspired by neighbors' solutions, leading to larger performance variance. As we might expect, this effect seems largest with the least-similar bot which was sharing the most unique, variance-increasing ideas.

This result also seems inconsistent with prior literature demonstrating that social following tends to reduce group variance (e.g., Salganik, Dodds, and Watts, 2006). However, as we explained above in response to another of the reviewer's comments, our experimental design required participants to discover options (nouns) on their own, probably resulting in the different data pattern from previous research.

We have incorporated the above plot as Extended Data Fig. 10b in the revised manuscript.

Reviewer #4's comments

The authors present a large-scale controlled experiment on the impact of bots on collective innovation in networks. In a 5x5 design of in total of 25 conditions, the impact of three different bots on the collective performance is investigated. The bot has two main functions, a) aggregate information in their local neighborhood and b) spread information across the network. The experimental design is intriguing. In particular, using word2vec as the backdrop for a fitness landscape is an exciting innovation. Yet, the

authors fail to formulate a clear hypothesis in the main manuscript regarding the effects of the bots. Furthermore, the results are inconclusive. The authors claim that bots in general and the “most-bot” specifically enhance creative discovery of human groups. Unfortunately, this claim does not seem to be supported by the data. The main concern here is not that results might be negative or inconclusive, but a lack of a strategy to mitigate false discoveries, and a misleading presentation of results. For this reason, I suggest revising the results section, potentially following the hypothesis outlined in the pre-registration.

We are grateful to learn that the reviewer positively evaluated our integration of word2vec into a controlled experimental design. We appreciate the reviewer’s suggestion regarding the presentation of the results, which we address point by point, as explained below.

The authors are addressing very relevant questions, regarding the impact of artificial agents on cultural processes with a focus on instant communication and aggregation. I would have liked to read a hypothesis of how these effects influence collective innovation. Unfortunately, although hypotheses are stated in the preregistration these are not picked up in the manuscript. Thus it is not clear if the presented analysis is hypothesis-driven. For an explorative analysis of 16 interaction terms, I would have expected some discussion on the danger of false discoveries and would have hoped for a related mitigation strategy.

There are two points raised by the reviewer here.

First, we apologize for the unclear description about whether the analyses conducted in the original manuscript have been preregistered or not. To clarify, we did indeed preregister our hypotheses regarding (1) the effects of the bot conditions and landscapes on group performance (Fig. 2) and (2) the quality of nouns shared by each type of the bots (Fig. 3a). In the revised manuscript, we added descriptions in the Method section as follows:

(lines 590-592)

The analyses presented in relation to Fig. 2 and Fig. 3a were preregistered analyses. Other analyses were not preregistered and thus were exploratory.

Second, we fully understand the reviewer’s concern about false discoveries. We agree that discussing the danger of false positive is needed regardless of our preregistration.

Accordingly, we have added a discussion on it and mitigated the possibility that readers interpret the findings too strongly, as follows:

(lines 567-574; Note that citations were represented by numbers in the main text)
While we adopted a mixed design (the decoy landscapes were between-subject treatments, while the bot conditions were within-subject treatments) to increase statistical power, and studied a large number of people and groups, we were still constrained by our sample size. Accordingly, we should note the danger of false discoveries (Benjamini and Hochberg, 1995; Simmons, Nelson, and Simonsohn, 2011), particularly for the exploratory analyses. For example, as reported, the effects of the interaction between the bot treatment and the landscape were not statistically significant when we examined the best solution in each game (Extended Data Fig. 5) although they were with respect to the average solution (**Fig. 2**).

There are important contradictions in the manuscript. In the abstract, one of the main presented findings is that the “most bot” enhances the exploration compared to groups without bots (27-29) and that bots, in general, enhance the capacity for creative discovery in human groups (31-32) [also: 469-471]. This is in contradiction to the first finding in the result section, which states that “the addition of any kind of bot [...] did not yield meaningful main effects”. Specifically for the most bot, Figure 2.b suggests that the larger part of the posterior density is below the zero line, suggesting, if anything, an inverse relationship.

We apologize for the unclear presentation of the findings. The reviewer is correct in stating that the performance increase provided by the most-similar bot was *conditional* in our study. To clarify this point, we adopted the reviewer’s description in the next comment and carefully revised the Abstract as follows:

Old manuscript

(lines 28-32)

Then we show that bots that share the most similar noun from among their neighbors’ proposals enhance this effect. This is the case even when we make the problem harder by adding decoys to the semantic space in order to disrupt individuals’ search patterns. Simple autonomous agents with interpretable behavior enhanced the capacity for creative discovery in human groups.

Revised manuscript

(lines 29-32)

Then we show that when bots that share the most similar noun operating in groups facing a semantic space that is relatively easy to navigate, group performance is superior. Simple autonomous agents with interpretable behavior can affect the capacity for creative discovery of human groups.

The Discussion section has been revised as follows:

Old manuscript

(lines 469-471)

The evidence presented here suggests that adding simple bots, acting with limited (if any) knowledge, but manipulating the sharing of ideas offered by human participants themselves, can facilitate creativity in human groups.

Revised manuscript

(lines 576-579)

The evidence presented here suggests that adding simple bots, acting with limited (if any) knowledge, but manipulating the sharing of ideas offered by human participants themselves within broader groups of humans, may enhance the creativity of human groups in certain circumstances.

The authors then analyze interactions between landscapes and bot types. Here they find a positive effect for “most: no decoy” and the “most: tall/narrow” interaction. Based on this, the authors conclude that “Groups with the most similar bot were found to be closer to the target noun in comparison to those in the no-bot condition, particularly in the no-decoy and the tall/narrow landscapes”. The first part of the sentence is in contradiction to the analysis of the main effects that clearly states that no difference between bot and non-bot conditions can be found (see previous paragraph). As the first part of the sentence is not supported, neither can the second part be supported based on the interaction alone. The interaction term does allow to conclude that for groups with the most bots on “easier” landscapes, there was a specific performance increase that cannot be explained by the bot type and the landscape alone. However, the interaction term alone does not allow to conclude a performance increase in comparison to the “no-bot condition” as the authors suggest. Such statement needs to also take into account the main effect of the bot type. In particular, as main effect (Most bot) appears to go in the opposite direction compared to the interaction (e.g. most: no decoy).

The reviewer's comment here also refers to the same issues as in the previous comment. We have revised the results section as follows:

Old manuscript

(lines 238-240)

Groups with the most-similar bot were found to be closer to the target noun in comparison to those in the no-bot condition, particularly in the no-decoy and the tall/narrow landscapes

Revised manuscript

(lines 251-253)

For groups with the most-similar bot and in some of the landscapes, we found a specific performance increase that cannot be explained by either the bot type or the landscape alone

Old manuscript

(lines 254-255)

To understand why the most-similar bot was effective at helping groups discover nouns with higher point values (in certain landscapes),

Revised manuscript

(line 270)

To understand why the most-similar bot had a positive effect (in certain landscapes)

We also clarified that there was no main effect of the bot types:

Revised manuscript

(lines 257-260)

There was no main effect of bot types (**Fig. 2b**), showing that the positive effect of the most-similar bot was not observed in comparison to the no-bot condition but was only seen as an interaction effect between bot function and landscape type.

Similarly, a few paragraphs later (257-261) the text suggests a pairwise comparison of the "most" and "least" bot across different landscape types. The statistical analysis of parameters listed in brackets seems however only to reflect the interaction between bot type and landscape. Here again, the main effect between both types of bots seems to

be not correctly accounted for.

We agree with the reviewer that the description of the results here should be revised as well.

Old manuscript

(lines 257-261)

Fig. 3a illustrates that the most-similar bot shared nouns that were closer to the target noun than the least-similar bot in the easier (no-decoy, short/narrow, and tall/narrow) landscapes ($\beta_{Least:no\ decoy} = -0.85$, 95% HDI = [-1.42, -0.25]; $\beta_{Least:short/narrow} = -0.81$, 95% HDI = [-1.45, -0.20]; $\beta_{Least:tall/narrow} = -0.81$, 95% HDI = [-1.40, -0.20]; using the tall/wide landscape as the reference).

Revised manuscript

(lines 272-277)

Fig. 3a illustrates that there was a sharing of higher quality nouns that cannot be explained by the bot type and the landscape alone for the most-similar bots employed with the “easier” (no-decoy, short/narrow, and tall/narrow) landscapes, compared to the least-similar bot ($\beta_{Least:no\ decoy} = -0.85$, 95% HDI = [-1.42, -0.25]; $\beta_{Least:short/narrow} = -0.81$, 95% HDI = [-1.45, -0.20]; $\beta_{Least:tall/narrow} = -0.81$, 95% HDI = [-1.40, -0.20]; using the tall/wide landscape as the reference).

We made it clear that the meaningful main effect was not observed in this analysis as well:

Revised manuscript

(line 281)

We did not observe statistically distinguishable main effects of the bot types.

Also in lines 296-301 main effects do not seem to be correctly accounted for.

We again thank the reviewer for pointing this out. We have added the explanation for the main effect to the revised manuscript as follows:

Revised manuscript

(lines 318-319)

We did not observe a meaningful main effect of the bot types, suggesting that none of the bot types led participants to the decoy when controlling for the landscape factor.

We hope all these careful revisions are seen as responsive.

The experimental design presented by the authors is an exciting innovation. It uses semantic similarity of nouns as extracted by contrastive learning (word2vec). The authors then construct an artificial fitness landscape on top of this semantic space. The word2vec space has two advantages, first it reflects in its structure a real-world concept space. As the author put, we can expect idea space to follow similar structures. Second, the word2vec space has been found to align with human associative similarity. This, as the authors have shown, allows naive participants to meaningfully explore the semantic space.

We are very glad that the reviewer found our experimental design with word2vec exciting; we also believe it to be an innovative aspect of our study.

Additional presented findings include:

- * Groups outperform individuals in searching on a fitness landscape
- * The most central items in the neighborhood of individual agents (here specifically bots) are of higher value than the least central one or a random one
- * The type of roughness informs the exploration pattern.
- * Individual performance only partially predicts performance in a group

The first two findings relate to the wisdom of the crowd effect. The third is novel to me. I would be curious to see a comparison with some rational learning strategy (see below). The last seems to relate to the notion of a collective intelligence factor c . I appreciate the efforts of the authors to report these findings in detail, however, the reader might benefit from stronger linking these findings to the existing literature on the wisdom of the crowd and collective intelligence.

The reviewer raised two points to be addressed here. The first concerns the comparison with certain rational learning strategies for the results presented in Fig. 4, which the reviewer detailed in comments just below; we addressed them there.

The second point is about linking the results of individual and collective performance to the idea of a collective intelligence factor c (Woolley et al., 2010) to clarify the relationship with the existing literature on collective intelligence. This is a great suggestion. We have incorporated additional descriptions as follows:

| (lines 409-414; Note that citations were represented by numbers in the main text)

Previous work has shown that collective intelligence can be a quality exhibited by a group, not always reducible to the individuals within it (Woolley et al., 2010). Our results suggest that there might be two traits that individuals possess to some degree: solo creativity and group creativity. Some individuals have high solo creativity and low group creativity, while others have high group creativity and low solo creativity. Together with previous literature, our results therefore help shed light on why individual traits do not always explain group performance.

The author analyses the correlation between the correlation of point values and the exploration distance (semantic distance between consequential nouns). They find this correlation to be larger on more narrow landscapes. The authors suggest that wider decoy peaks hinder cognitive alignment (~330). I would suggest exploring an alternative hypothesis. For instance, it might be rational for a Bayesian agent to explore more locally on a more pointy landscape when approaching a peak.

We appreciate the reviewer's feedback on an alternative hypothesis. Based on the Bayesian agent hypothesis, we expected that Bayesian rational participants would start to explore more locally in later rounds of a game, even in a wider landscape.

To check this possibility, we analyzed the exploration distance between each round (ranging from 1 to 24). The figure shows the declining trend in the semantic similarity in the wider landscapes. Contrary to our expectations, the result indicates that participants in the wider landscape did not explore locally in later rounds.

We agree with the reviewer that there is a possibility participants were Bayesian rational, but we did not obtain a supportive result for that hypothesis. Accordingly, we decided to leave the hypothesis and the analysis in this response letter.

In conclusion, while the experimental design is innovative and the execution is impressive, some of the main claims made by the authors are in my view not sufficiently supported. Correspondingly, I suggest a thorough revision of the result section. Code and data has not been provided and was correspondingly not reviewed. As a final remark, the work could more clearly demonstrate its novelty.

Thank you for the encouraging comment on our work. As explained above, we have carefully revised the manuscript so that we can communicate our findings more correctly and clearly. We believe the revision has strengthened our work's credibility and clarity.

Minor remarks:

Abstract: The last sentence seems to miss a connection.

According to the previous reviewer's comment, we have revised the Abstract. Here we are presenting the last two sentences of the Abstract:

Old manuscript

(lines 29-32)

This is the case even when we make the problem harder by adding decoys to the semantic space in order to disrupt individuals' search patterns. Simple autonomous agents with interpretable behavior enhanced the capacity for creative discovery in human groups.

Revised manuscript

(lines 29-32)

Then we show that when bots that share the most similar noun operating in groups facing a semantic space that is relatively easy to navigate, group performance is superior. Simple autonomous agents with interpretable behavior can affect the capacity for creative discovery of human groups.

Introduction: Although the impact of bots on cultural evolution appears to be the main focus of this research, bots are only mentioned in the very last two paragraphs. I suggest improving the link between the general remarks on cultural evolution and bots and clearly stating the motivation for the experimental design.

We agree with the reviewer that improving the connection between the context of cultural evolution and artificial bots makes the motivation for our work even clearer. We have revised the Introduction as described below:

(lines 45-46; Note that citations were represented by numbers in the main text)

Moreover, since simple artificial agents can alter group behavior in a variety of ways (Jung et al., 2015; Kim et al., 2013; Shirado & Christakis, 2017; Shirado & Christakis, 2020; Traeger et al., 2020), such agents might also affect the specifically creative capacity of groups.

281: Figure 3a shows that the most central word within the bot neighborhood is of higher value compared to the least or a random word for most landscapes. This, however, does not speak about whether this helped subjects. I suggest more careful wording.

Following the reviewer's suggestion, we revised the manuscript to use more careful wording as follows:

Old manuscript

(lines 265-266)

These results suggest that the most-similar bot was able to help subjects generate nouns that were similar to the target by propagating high-value nouns throughout a network.

Revised manuscript

(lines 283-285)

Although this evidence is indirect, these results suggest that the most-similar bot may have been able to help subjects generate nouns that were similar to the target by propagating high-value nouns throughout a network.

Old manuscript

(lines 270-272)

the bots amplify this ability by leveraging the wisdom of crowds, essentially averaging guesses and reducing noise.

Revised manuscript

(lines 289-290)

the bots may amplify this ability by leveraging the wisdom of crowds, essentially reducing noise.

274: I got a bit confused by the figure description of Figure 3. Does it show the mean of the raw data or estimates of model parameters?

We are sorry for the unclear description. The original Fig. 3 showed the mean of the raw data. We revised the Fig.3 to present the same data with box-and-whisker plots according to the Editor's comment. With the new plots, we have updated the figure description as follows:

(lines 305-306)

The plots show summary statistics of the raw data presented with box-and-whisker plots.

291-292: Something seems to be missing in this sentence, as the explanation seems to be incomplete.

To make the transition smoother, we revised the manuscript in a following way:

Old manuscript

(lines 290-292)

Our data indicate that the addition of the most-similar bot helps groups approach the target noun. However, it is conceivable that the bot would also cause subjects to become stuck in a local optimum simply by sharing nouns.

Revised manuscript

(lines 311-314)

Our data indicate that the addition of the most-similar bot helps groups approach the target noun, but it is conceivable that the bot could also lead groups towards the decoy noun by sharing nouns whose point values have been artificially inflated due to proximity with the target noun.

263: I was surprised to see two different HDI reported (90% and 95%) for different parameters

We revised the manuscript to report the same HDI (95%) for all the analyses. On some occasions, we also furthermore report the 90% HDI for the results that included zero within the 95% intervals in order to communicate nuanced results.

Old manuscript

(lines 241-242)

Similar trends were also shown in the short/narrow landscape ($\beta_{Most:short/narrow} = 0.44$; 90% HDI [0.00, 0.87]).

Revised manuscript

(lines 254-255)

Similar trends were also shown in the short/narrow landscape ($\beta_{Most:short/narrow} = 0.44$; 95% HDI [-0.08, 0.97]; 90% HDI [0.00, 0.87]).

Old manuscript

(lines 261-263)

Likewise, the most-similar bots meaningfully outperform the random bots in the no-decoy and tall/narrow landscapes ($\beta_{Random:no\ decoy} = -0.47$, 90% HDI = [-0.98, 0.01]; $\beta_{Random:tall/narrow} = -0.85$, 95% HDI = [-1.44, -0.23]).

Revised manuscript

(lines 277-281)

Likewise, the most-similar bots meaningfully outperform the random bots in the tall/narrow landscape ($\beta_{Random:tall/narrow} = -0.85$, 95% HDI [-1.44, -0.23]). A similar trend was also observed in in the no-decoy landscape ($\beta_{Random:no\ decoy} = -0.47$, 95% HDI [-1.04, 0.14]; 90% HDI [-0.98, 0.01]).

340: Could it be that participants in the group just get inspired and use nouns from other participants as a starting point for their brainstorming? Simple heuristics without any world model might be able to explain the same behavior. I suggest being careful with concluding that participants form some kind of model of the fitness landscape.

This is a very good point. We agree that both forming a model of the landscape and using simple heuristics can simultaneously be the case. We revised the Discussion by adopting the reviewer's description as follows:

(lines 475-481)

The reason participants in the solo condition did not perform *better* than those in the group conditions even when we looked at the best guesses might be explained by the fact that participants were required to come up with nouns and were not provided with options to explore in the present experiment. Thus, some of the positive performance shown in the group conditions should be attributed to simple heuristics that use nouns from other participants as a starting point for brainstorming rather than forming a mental model of the fitness landscape.

REVIEWERS' COMMENTS

Reviewer #1 (Remarks to the Author):

The authors have thoroughly responded to the questions and feedbacks raised by my review as well as those raised by other three reviewers. Now I see that the MS has been much improved, with results and analyses being convincing. I appreciate the efforts they put in revision. I believe the MS can be published in the current version. I would like to say congratulations to the authors.

Warmest wishes,
Wataru Toyokawa

Reviewer #2 (Remarks to the Author):

Overall, I am happy with the revisions made by the authors for this version of the manuscript. I feel that they adequately addressed the critiques that I raised for the previous version of the manuscript, and I have no additional feedback to provide.

Sincerely,
Ryan L. Boyd

Reviewer #3 (Remarks to the Author):

The revision responses are sound and I'm happy to endorse the publication.

Regarding Hills, T. (2023), the correct citation is as follow:

Hills, T. (2023). Group problem solving: Harnessing the Wisdom of the Crowds. PsyArxiv Prepr.
<https://doi.org/10.31234/osf.io/eyrzg>

Reviewer #4 (Remarks to the Author):

I appreciate the considerable efforts by the authors to address the concerns raised in the previous review. The revisions have adequately addressed all the major concerns mentioned in my initial review, resulting in a significant improvement in the manuscript. In particular, the clarity in stating the scope of the findings has been considerably enhanced. I appreciate the authors being open to the provided suggestings for considering alternative interpretations of the reported findings.

I am looking forward to seeing this work being published in Nature Communications soon.

Reviewer #4 (Remarks on code availability):

The code provided is well-structured and appears to contain all necessary information for reproduction.

Reviewer #1's comments

The authors have thoroughly responded to the questions and feedbacks raised by my review as well as those raised by other three reviewers. Now I see that the MS has been much improved, with results and analyses being convincing. I appreciate the efforts they put in revision. I believe the MS can be published in the current version. I would like to say congratulations to the authors.

Warmest wishes,
Wataru Toyokawa

We are grateful to Reviewer #1 for these kind words.

Reviewer #2's comments

Overall, I am happy with the revisions made by the authors for this version of the manuscript. I feel that they adequately addressed the critiques that I raised for the previous version of the manuscript, and I have no additional feedback to provide.

Sincerely,
Ryan L. Boyd

We are grateful to Reviewer #2 for these kind words. We believe that incorporating a clearer explanation of the relationship between the motivation, experimental design, and theoretical framework, based on previous comments from Reviewer #2, has made the manuscript even better.

Reviewer #3's comments

The revision responses are sound and I'm happy to endorse the publication.

Regarding Hills, T. (2023), the correct citation is as follow:

Hills, T. (2023). Group problem solving: Harnessing the Wisdom of the Crowds. PsyArxiv Prepr. <https://doi.org/10.31234/osf.io/eyrzig>

Thank you for these kind words. We have also fixed this reference.

Reviewer #4's comments

Reviewer #4 (Remarks to the Author):

I appreciate the considerable efforts by the authors to address the concerns raised in the previous review. The revisions have adequately addressed all the major concerns mentioned in my initial review, resulting in a significant improvement in the manuscript. In particular, the clarity in stating the scope of the findings has been considerably enhanced. I appreciate the authors being open to the provided suggestings for considering alternative interpretations of the reported findings.

I am looking forward to seeing this work being published in Nature Communications soon.

Reviewer #4 (Remarks on code availability):

The code provided is well-structured and appears to contain all necessary information for reproduction.

We are grateful to Reviewer #4 for these kind words. We particularly thank the reviewer for suggesting an important alternative interpretation of the results, which we added.